# CHANGE POINT DETECTION VIA VARIATIONAL TIME-VARYING HIDDEN MARKOV MODEL

## ABSTRACT

The task of modeling time series data that exhibit sudden regime shifts has been an enduring focus of research due to its inherent complexity. Among the various strategies to tackle this issue, the Hidden Markov Model (HMM) has been extensively investigated, which captures the regime changes by modeling the transition between latent states. Despite its popularity, the HMM-based methodology carries certain limitations, including specific distribution assumptions and its computational intensity for inference and learning, particularly when the number of change points is unidentified. In this work, we propose a novel approach that models the location of change points and introduce the **TV-HMM**, a variant of the Hidden Markov Model incorporating the time-varying location transition matrix. Based on the novel modeling scheme, we propose an associated variational EM algorithm that simultaneously detects the locations and the number of change points, together with inferring the posterior distributions of regime parameters. In contrast to previous approaches, the proposed method exhibits robustness against the misspecification of change point numbers and can be augmented with stochastic approximation techniques to effectively mitigate the computational burden. Furthermore, we establish the statistical consistency of the change point location estimation under the Gaussian likelihood assumption. We also generalize the parametric likelihood function using the Maximum Mean Discrepancy (MMD) and propose the semi-parametric **TV-HMM** that is free of distribution assumptions. A series of experiments validate the theoretical convergence rate and demonstrate our estimation accuracy in terms of Rand index and MSE.

## 1 INTRODUCTION

One of the fundamental tasks in signal processing and time series analysis is identifying and analyzing a complex system with temporal evolution. The states of systems are measured over time by a sequence of observations. Evaluating the locations of abrupt distributional changes within the sequence is commonly known as the Change Point Detection (CPD) problem. Practically, many applications require solving the CPD problem, where the proposed methods are helpful for subsequent analysis of the sequence characteristics, such as gait analysis (Lee & Grimson, 2002), anomaly detection (Liu et al., 2018), biological diagnostics (Gardner et al., 2006), financial analysis (Andreou & Ghysels, 2002), and more.

In this paper, we focus on offline change point detection methods (Truong et al., 2020), which analyze and operate the complete dataset retrospectively. Compared to the online CPD methods (Adams & MacKay, 2007; Chang et al., 2019), these methods are better suited for complex modeling and they have access to the entire observations, which enables higher detection accuracy and a more comprehensive understanding of the overall patterns, trends, and characteristics of the regimes between the adjacent change points.

There is rich literature related to the offline change point detection problem. The early work can be traced back to the 1950s, which focuses on detecting the mean value changes of independent and identically distributed (i.i.d) Gaussian random variables (Page, 1955). From the methodology perspective, Pein et al. (2017) detects change points based on ubiquitous maximum likelihood estimation. With the piecewise linear model assumption, Bai & Perron (1998) minimizes the squared and absolute cost function on the observed sequence and the parameters. Harchaoui & Cappé (2007)

detect the change point by minimizing the kernel distance of observations in reproducing kernel Hilbert space while Zou et al. (2014) uses the empirical distribution divergence measurement. When it comes to the Bayesian approaches, Barry & Hartigan (1993); Park & Dunson (2010); Müller et al. (2011) develop the product partition model(PPM) for offline CPD, and Chib (1998) introduces a Hidden Markov Model (HMM) and determines the latent change point state by the Markov Chain Monte Carlo (MCMC) algorithm. Pesaran et al. (2006) introduces a hierarchical structure on HMM where parameters follow certain common meta-distributions. Assuming regime durations have a Poisson distribution, Koop & Potter (2007) develops a time-varying parameter model with hierarchical prior distributions to detect change points. Additionally, Ko et al. (2015) combines the Dirichlet process with HMM to estimate the latent state without prior specification of the number of states. The comprehensive review of offline change point methods can be found in (Truong et al., 2020).

However, the effectiveness of these methods can be influenced by various hyper-parameters, e.g., the number of change points, significance level, or penalty coefficients. Killick et al. (2012) adapts the CPD algorithm with a linear penalty on the number of change points. Determining the optimal values for these parameters may require specialized knowledge or additional evaluation criteria (Burnham & Anderson, 2004). Although some non-parametric Bayesian models (Ko et al., 2015; Peluso et al., 2019) do not require a predetermined number of change points, they often involve computationally intensive processes, such as MCMC sampling, to obtain posterior distributions for the entire dataset. Furthermore, previous studies on Bayesian CPD have mainly focused on algorithmic design and lack strong theoretical guarantees. The convergence rate and performance of these methods may vary depending on the specific problem and settings. Additionally, many CPD methods rely on parametric distributions, often assuming each observation to be normally distributed in order to detect changes in mean and variance. While these assumptions offer advantages in terms of interpretability and inference efficiency, it is still preferable to have a CPD method that is not limited by the likelihood assumption, as it would be more robust against model misspecification and outliers. Therefore, these limitations make these methods less practical for real-world applications and datasets.

In order to overcome the challenges of hyperparameter selection and computational burden, we propose the Time-Varying Hidden Markov Model (**TV-HMM**). Concisely, our contributions are as follows:

1) **TV-HMM** models the locations of change points with the time-varying Markov chain. Its transition matrix takes into account the size of the sequence length, encompassing all possible locations. The adaptive updating of the transition matrix for each change point allows for more efficient change point detection without prior knowledge of the number of change points.

2) We develop a variational EM algorithm that can endogenously determine the necessary number of change points from the observed data. The algorithm leverages stochastic approximation by chronologically sampling an observation subset. This reduces the computational cost compared to MCMC-based inference. Our theoretical analysis demonstrates the statistical consistency of our method in detecting change point locations.

3) To validate our theoretical results, we conduct numerical experiments and evaluate the performance of our proposed method on both simulated and real-world data. These experiments demonstrate the effectiveness and robustness of our approach.

4) We extend the parametric method to the semi-parametric **TV-HMM** that alleviates the assumption on parametric distribution by using Maximum Mean Discrepancy (MMD) for likelihood measurement. We introduce a new learning objective, MMD-ELBO, and train the model through re-parameterization trick (Kingma et al., 2015). Our experiments show promising performance on non-Gaussian datasets without incorporating distributional knowledge.

## 2 TIME-VARYING HIDDEN MARKOV MODEL AND LOCATION TRANSITION

Given the observed $D$-dimensional sequence $\mathbf{Y} = \{y_1, \ldots, y_N\}$ with $y_n \in \mathbb{R}^D$, our goal is to detect all $K$ change point $\{\tau_k\}_{k=1}^K$, with each $\tau_k \in \{1, \ldots, N\}$ and estimate the distribution of each regime. There are extensive works with different settings of CPD, such as piecewise i.i.d assumption (Matteson & James, 2014; Li et al., 2015), autoregressive assumption (Yamanishi & Takeuchi, 2002), and others (Kawahara et al., 2007). In this work, we illustrate our method using

the common piecewise i.i.d setting, such that $y_n$ is independently sampled from a distribution $\mathcal{P}_k$ for $\tau_{k-1} \leq n \leq \tau_k$, with $\tau_0 = 1$ and $\tau_{K+1} = N$.

## 2.1 Time-Varying Hidden Markov Model: A Parametric Case

We encode the change point location by a one-hot random variable $\mathbf{t}_k \in \mathbb{R}^N$. Since the $k$-th change point should be always no earlier than the $k-1$-th, the stochastic process $\{\mathbf{t}_1, ..., \mathbf{t}_K\}$ is a left-to-right Markov chain with a upper triangular transition matrix. Denoting $\mathbf{t}_k(i)$ as the $i$-th element of the vector, the joint distribution of $\{\mathbf{t}_1, \ldots, \mathbf{t}_K\}$ as well as the transition probability matrix between $\mathbf{t}_{k-1}$ and $\mathbf{t}_k$ are given by:

$$p(\mathbf{t}_1; \Pi_1)p(\mathbf{t}_2|\mathbf{t}_1; \Pi_2)\ldots p(\mathbf{t}_K|\mathbf{t}_{K-1}; \Pi_K) = \prod_{n=1}^{N} \pi_{1,n}^{\mathbf{t}_1(n)} \prod_{k=2}^{K} \left[ \prod_{n=1}^{N} \prod_{m=1}^{N} \pi_{k,n,m}^{\mathbf{t}_k(n) \times \mathbf{t}_{k-1}(m)} \right],$$

$$\text{with} \quad \Pi_k := \begin{bmatrix} \pi_{k,1,1} & \pi_{k,1,2} & \cdots & \pi_{k,1,N-1} & \pi_{k,1,N} \\ 0 & \pi_{k,2,2} & \cdots & \pi_{k,2,N-1} & \pi_{k,2,N} \\ \vdots & \vdots & \ddots & \vdots & \vdots \\ 0 & 0 & \cdots & \pi_{k,N-1,N-1} & \pi_{k,N-1,N} \\ 0 & 0 & \cdots & 0 & \pi_{k,N,N} \end{bmatrix},$$

where each element $\pi_{k,i,j}$ represents the prior probability coefficient that $k$-th regime starts at time step $i$ and ends at $j$. Note that the previous hidden Markov models(Chib, 1998; Ko et al., 2015) consider a restricted transition matrix whose size is proportional to the state number $K$. The Markov chain in these methods experiences $N$-step transitions along the sequence. On the other hand, our modeling scheme allows the transition probability matrix $\Pi_k$ to evolve over time and only computes $K$-step transitions to improve the inference efficiency.

Under the parametric case, the distribution shift between the adjacent regimes is reduced to the change of parameter values. We treat the regime parameters $(\theta_1, \ldots, \theta_{K+1})$ as random variables and introduce $K + 1$ prior distributions $\{p(\theta_k; \alpha_k)\}_{k=1}^{K+1}$, where $\alpha_k$ denotes all hyperparameters for $k$-th regime . For illustration purposes, we consider the Gaussian likelihood case with mean and precision, where $\theta_k = \{u_k, \Lambda_k\}$ and the conjugate normal-Wishart prior. Given the location indicators $(\mathbf{t}_k, \mathbf{t}_{k-1})$, the random variable $\mathbf{Y}_k$ represents the observations set within the corresponding regime. Under the piecewise i.i.d assumption, the likelihood of the $k$-th regime and prior distributions is given by:

$$p(\mathbf{Y}_k|\mathbf{t}_k, \mathbf{t}_{k-1}, \theta_k) = \prod_{i=1}^{N} \prod_{j=i}^{N} \left[ \prod_{t=j}^{i} \mathcal{N}(y_t \mid u_k, \Lambda_k) \right]^{\mathbf{t}_{k-1}(i) \times \mathbf{t}_k(j)}, \tag{1}$$

$$u_k \sim \mathcal{N}(0, \beta^{-1}\mathbf{I}), \quad \Lambda_k \sim \mathcal{W}(v_0, V_0),$$

where $\mathcal{W}(\cdot)$ denotes the Wishart distribution. In our model specification, $\pi_{k,i,j}$ can be learned directly from data by optimizing with respect to marginal data likelihood. This probability reflects the relevance of time interval $[i, j]$ with the true regime $[\tau_k, \tau_{k-1}]$. A similar idea has been applied in the hyperparameters learning of Gaussian process (Rasmussen et al., 2006). In our model, since practically the value of $K$ is unknown, automatic model selection can be performed by learning these probabilities for each $\mathbf{t}_k$. If the corresponding diagonal elements $\pi_{k,i,i}$ converge to 1, indicating the time length of $k$-th regime is zero, then the unnecessary change points can be removed from the model specification. In **Section** 3.1, we visualize the value of converged $\Pi_k$ from numerical simulations and illustrate all significant spots concentrating on the true locations and the diagonal.

## 2.2 Inference via Variational EM Algorithm

Denoting the set of all latent variables as $\boldsymbol{\xi} = \{\{\mathbf{t}_k\}_{k=1}^{K}, \{\theta_k\}_{k=1}^{K+1}\}$, **TV-HMM** detects the locations and number of change points by inferring the posterior distribution of $\boldsymbol{\xi}$ and learning the transition probability $\Pi_k$. In the Bayesian literature, Neal (2012) introduces the automatic relevance determination (ARD) procedure for neural network learning. The idea is that optimizing the continuous hyperparameters with respect to marginal log-likelihood provably leads to consistent model selection and obeys Occam's razor phenomenon (Ghosal et al., 2008; Yang & Pati, 2017). However,

---

**Algorithm 1** Variational EM algorithm for Time-Varying Hidden Markov Model

---

**Input:** The observed sequence: $\mathbf{Y}$; The initial number of change points: $\tilde{K}$; Maximum iteration: $I$; Size of sampling subset: $S$; Step size: $\eta$;

**Output:** Variational distributions $\{Q(\theta_k)\}_{k=1}^{\tilde{K}+1}$; Marginal probability of CP locations $\{Q(\mathbf{t}_k)\}_{k=1}^{\tilde{K}}$;

1: Initialization of variational expectation of $\{Q(\theta_k)\}_{k=1}^{\tilde{K}+1}$;
2: **for** $1 \leq i_1 \leq I$ **do**
3:    **Random sampling $S$ data point and collect the retrospective order set $\Omega$ ;**
4:    **E-Step:**
5:    Update variational distributions $\{Q^S(\mathbf{t}_k)\}_{k=1}^{\tilde{K}}, \{Q^S(\mathbf{t}_k, \mathbf{t}_{k-1})\}_{k=2}^{\tilde{K}}$ by **Equation** 3 based on sampled $S$ observations, with;

$$Q^S(\mathbf{t}_k(n) = 1 \mid \mathbf{t}_{k-1}(m) = 1) = \begin{cases} Q^S(\mathbf{t}_k(n) = 1 \mid \mathbf{t}_{k-1}(m) = 1) & \text{if } m, n \in \Omega, \\ 0 & \text{otherwise} \end{cases}$$

6:    Re-estimate $\{Q(\theta_{\mathbf{k}})\}_{k=1}^{\tilde{K}+1}$ using **Equation** 2 based on sampled $S$ observations;
7:    **M-Step:**
8:    Set new prior by $\pi_{k,m,n} \leftarrow \pi_{k,m,n} + \eta \cdot Q^S(\mathbf{t}_k(n) = 1 \mid \mathbf{t}_{k-1}(m) = 1)$;
9: **end for**

---

direct marginal likelihood maximization is intractable since it involves the integral over all latent variables. EM algorithm provides a solution where we relax the marginal likelihood function with its lower bound. Denoting the hyperparameters set $\mathbf{\Pi} = \{\Pi_k\}_{k=1}^K$ and $\boldsymbol{\alpha} = \{\alpha_k\}_{k=1}^{K+1}$, we have

$$\text{E-Step. } \mathcal{L}\left(\mathbf{\Pi} \mid \mathbf{\Pi}^{old}\right) = \mathrm{E}_{\boldsymbol{\xi}\mid\mathbf{Y};\mathbf{\Pi}^{old},\boldsymbol{\alpha}}[\log p(\mathbf{Y}, \boldsymbol{\xi}; \mathbf{\Pi}, \boldsymbol{\alpha})],$$

$$\text{M-Step. } \hat{\mathbf{\Pi}} = \arg\max_{\mathbf{\Pi}} \mathcal{L}\left(\mathbf{\Pi} \mid \mathbf{\Pi}^{old}\right).$$

Although the EM algorithm seems feasible, the E-Step requires evaluating the true posterior $p(\boldsymbol{\xi} \mid \mathbf{Y}; \mathbf{\Pi}, \boldsymbol{\alpha})$, which has no analytical form. Here we further leverage variational approximation and introduce a tractable variational distribution $Q$ as an approximator of the true posterior under KL divergence. By maximizing the evidence lower bound (ELBO), we minimize the KL divergence between $Q$ and the true posterior distribution (Blei et al., 2017). Under common mean-field assumption where variational distributions can be independently factorized, we can obtain explicit solutions of optimal approximator $Q^*$:

$$Q^*(\theta_k) \propto \exp\left(\mathrm{E}_{Q(\mathbf{t}_k, \mathbf{t}_{k-1})}\left[\log p(\mathbf{Y}_k, \mathbf{t}_k, \theta_k \mid \mathbf{t}_{k-1}; \alpha_k, \Pi_k)\right]\right),$$

$$Q^*(\mathbf{t}_1, ..., \mathbf{t}_K) \propto \prod_{k=1}^{K+1} \exp\left(\mathrm{E}_{Q(\theta_k)} \ln p(\mathbf{Y}_k, \mathbf{t}_k, \theta_k \mid \mathbf{t}_{k-1}; \Pi_k, \alpha_k)\right). \tag{2}$$

Noted that the solution in **Equation** 2 is a joint distribution of $\{\mathbf{t}_1, \ldots, \mathbf{t}_K\}$. However, the primary interest of location detection is marginal distributions $Q(\mathbf{t}_k)$, and $Q(\mathbf{t}_k, \mathbf{t}_{k-1})$ for $Q^*(\theta_k)$ inference. To obtain these quantities, we propose a recursive message-passing procedure based on the sum-product algorithm. The marginalization is achieved by passing real-valued message functions between the latent variables $\mathbf{t}_k$, which are denoted by: $\mu_{\rightarrow \mathbf{t}_k}, \mu_{\mathbf{t}_k \leftarrow} \in \mathbb{R}^N$. These two functions represent the information flow that propagates front and back from subsequent variables:

$$Q(\mathbf{t}_k(n) = 1) \propto \mu_{\rightarrow \mathbf{t}_k}(n) \cdot \mu_{t_k \leftarrow}(n), \quad Q(\mathbf{t}_{k-1}(m) = 1, \mathbf{t}_k(n) = 1) \propto$$

$$\mu_{\rightarrow \mathbf{t}_{k-1}}(m) \cdot \pi_{i,m,n} \cdot \exp\left(\mathrm{E}_{Q(\theta_k)} \ln p(\mathbf{Y}_k, \mathbf{t}_k, \theta_k \mid \mathbf{t}_{k-1}; \Pi_k, \alpha_k)\right) \cdot \mu_{t_k \leftarrow}(n),$$

where the recursive formula of message passing is given by:

$$\mu_{\rightarrow \mathbf{t}_k}(n) = \sum_{m=1}^n \left\{\mu_{\rightarrow \mathbf{t}_{i-1}}(m) \cdot \pi_{k,m,n} \cdot \exp\left[\mathrm{E}_{Q(\theta_k)} \ln p(\mathbf{Y}_k \mid \theta_k, \mathbf{t}_k(n) = 1, \mathbf{t}_{k-1}(m) = 1)\right]\right\},$$

$$\mu_{\mathbf{t}_{k-1} \leftarrow}(m) = \sum_{n=m}^N \left\{\mu_{\mathbf{t}_k \leftarrow}(n) \cdot \pi_{k,m,n} \cdot \exp\left[\mathrm{E}_{Q(\theta_k)} \ln p(\mathbf{Y}_k \mid \theta_k, \mathbf{t}_{k-1}(m) = 1, \mathbf{t}_k(n) = 1)\right]\right\}.$$

$$\tag{3}$$

Given the initial $\mu_{\to \mathbf{t}_1}$ and $\mu_{\mathbf{t}_K \leftarrow}$, each message flow can be iteratively evaluated. For the Gaussian example of **Equation** 1, the detailed expressions of **Equation** 2 and 3 are given in the Appendix B. After updating all variational distributions by taking one-step coordinate gradient ascent, we optimize hyperparameters $\mathbf{\Pi}$ in M-step. By alternating between E and M steps, we simultaneously detect the change point locations and estimate the parameters of each regime using the maximum a posteriori probability (MAP) of variational distributions:

$$\hat{\tau}_k = \arg\max_{\tau_k} Q\left(\mathbf{t}_k(\tau_k) = 1\right), \quad \hat{\theta}_k = \arg\max_{\theta_k} Q\left(\theta_k\right).$$

The computation complexity of each iteration is $\mathcal{O}(KN^2)$. When the length of the sequence grows, the convergence speed and memory usage become inhibited. To relieve the computational burden, we randomly sample a subset of observations chronologically in each iteration. The subset has a fixed length $S$, which is much smaller than the number of observations $S \ll N$. A local estimator with this subset is established under stochastic approximation that enjoys less computational complexity and guarantees convergence to global optimal (Robbins & Monro, 1951). In our simulations, the proposed procedure usually converges or reaches the predefined iteration limit within 30 iterations. Thus, we successfully reduce the computational cost of each EM step to $\mathcal{O}(KS^2)$ and improve the computational efficiency. The complete procedure is summarized in **Algorithm** 1.

Practically, the unknown prior knowledge of $K$ could be learned from data using 'ARD'. If we initialize our method using a Markov chain $[\mathbf{t}_1, ..., \mathbf{t}_{\tilde{K}}]$ with $\tilde{K} > K$. As the algorithm progresses, the learned transition matrix $\mathbf{\Pi}$ reveals the probability of each location transition, and the estimated locations $\{\hat{\tau}_k\}_{k=1}^{\tilde{K}}$ are clustered together. Some of the successive change points will gradually converge to the same location, e.g. $\hat{\tau}_{L_1} = \hat{\tau}_{L_1+1} = ... = \hat{\tau}_{L_1+l}$ for some integer $l$. Therefore, the redundant regimes will vanish during the EM iteration and there are only $K$ unique locations remaining after convergence.

## 2.3 THEORETICAL ANALYSIS

In this section, we provide a statistical analysis of how **TV-HMM** estimates the change point locations and numbers. We list the necessary notations and assumptions under which our theoretical result is established:

**A1**: For fixed constants $T, K, D$, the underlying sequence on time interval $[0, T]$ consists of $K$ change points $0 < T_1 < ... < T_K < T_{K+1} = T$ and the random function $\mathrm{y}(t) : \mathbb{R} \to \mathbb{R}^D$ represents the sample drawn from $\mathcal{N}(y \mid u_k, \Lambda_k)$ if $T_{k-1} < t < T_k$.

**A2**: For any time interval $[m, n] \subseteq [0, T]$, the number of observations within this interval equals $\mathcal{O}(N^{\frac{n-m}{T}})$.

**A3**: The algorithm initializes $\tilde{K} = M_{K+1} - 1$ change points. Each corresponds to an equal-distance segment $[t_{i-1}, t_i]$, such that $t_{i+1} - t_i = \frac{T}{M_{K+1}}$. We can further categorize $\{\mathbf{t}_i\}_{i=1}^{M_{K+1}-1}$ into two subsets:

- Any $\mathbf{t}_i$ with $i \in \{M_1, ..., M_K\}$ denotes the junction points, e.g. there is a true change point located within the interval $[t_{i-1}, t_i]$ and $\mathrm{y}(t)$ within the interval does not identically distributes.
- For $k = 1, ..., K + 1$, any $\mathbf{t}_i$ with $i \in \{M_{k-1} + 1, ..., M_k - 1\}$ denotes the non-junction index, such that every $\mathrm{y}(t)$ within the interval comes from the same distribution.

Then we can show our method leads to a provable selection consistency of change point locations:

**Theorem 1** (Location Consistency). *Assuming assumption A1-A3 hold, the marginal probability $Q\left(\mathbf{t}_i(n) = 1\right)$ consistently estimates the location of the change point with the maximum exponential rate of $N$:*

- *For non-junction points $\mathbf{t}_i$ with $i \in \{M_{k-1} + 1, ..., M_k - 1\}$:*

$$Q\left(\mathbf{t}_i(n) = 1\right) = \begin{cases} 1 & \text{if } n = T_k; \\ \mathcal{O}(N^{-\frac{n-T_k}{T}}) & \text{if } n \in [T_{k-1}, T_k); \\ \mathcal{O}(\exp(-N^{\frac{\min\left\{|n-T_k|, |n-T_{k-1}|\right\}}{T}})) & \text{if } n \notin [T_{k-1}, T_k]. \end{cases}$$

- *For junction points $\mathbf{t}_i$ with $i \in \{M_k\}_{k=1}^{K}$:*

$$Q\left(\mathbf{t}_i(n) = 1\right) = \begin{cases} 1 & \text{if } n = T_k; \\ \mathcal{O}(\exp(-N^{\frac{|n-T_k|}{T}})) & \text{otherwise.} \end{cases}$$

**Remark**. The assumption **A3** guarantees that each $Q(\theta_k)$ is initialized using the characteristic (e.g. mean and variance for the Gaussian case) of equal distance segments $[t_{k-1}, t_k]$, which is depicted with a box in Figure 1. Then **Theorem** 1 indicates these segments determine convergence rates of probabilities $Q(\mathbf{t}_k)$, e.g. if the segment contains a true change point $T_k$, $\mathbf{t}_k$ is a junction point and its $Q(\mathbf{t}_k(n))$ would converge to 1 for $n = T_k$ at the exponential rate of $N$. On the other hand, non-junction points whose initial segments are identically distributed with the true regime will also converge at the rate up to the exponential of $N$. Thus, as $N \to \infty$, the MAP estimations of $\{\hat{\tau}_k\}_{k=1}^{M_{K+1}-1}$ become an unduplicated set $\{T_k\}_{k=1}^{K}$ and can drop those segments whose length are $0$.

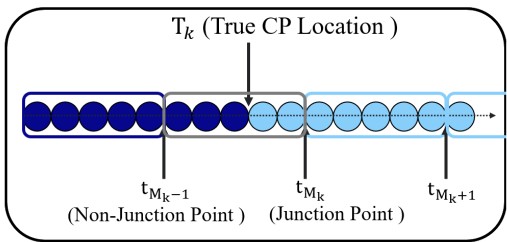

Figure 1: The schematic diagram of initialization in **Algorithm** 1. The initialized change points can be categorized into (non)-junction points based on the true location $T_k$.

## 3 SYNTHESIS DATA ANALYSIS

In this section, we evaluate our method on various simulations and real data. We first conduct numerical experiments to provide evidence for our theoretical result. Then we compare the performance of **TV-HMM** with that of other offline CPD methods in both simulated data and the real-world application. These results show the effectiveness and robustness of our method in terms of location detection and parameter estimation. Throughout the experiments, we evenly divide the sequence into $\tilde{K}$ segments to fulfill **A3** in **Section** 2.3. The details about initialization and hyperparameters setting are included in the Appendix D.1

### 3.1 IN-DEPTH ANALYSIS OF THEOREM 1

To analyze the theoretical results with controlled experiments, we consider a normal mean-variance shift model, which is also studied in (Yamanishi & Takeuchi, 2002; Matteson & James, 2014). The performance of CPD is measured by mean absolute error (**MAE**). For true change point location $\{l_1, l_2, \ldots\}$ and estimated $\{\hat{l}_1, \hat{l}_2, \ldots\}$, $\mathbf{MAE} = \frac{1}{N} \sum_j \min_i |\hat{l}_j - l_i|$, which measures the sum of absolute distances of each estimated location with its closest true location.

We first investigate the change of convergence rate by varying the value of $N$ and the results are summarized in Figure 2. The top left plot (a) shows the small value of $N$ results in fluctuations of the estimated number of change points; as the size of observations increases, the estimated number remains steady at the true value $4$. Similarly, the performance of parameters estimation is in the bottom left plot (b), indicating the estimation error rapidly decreases as the length of sequences grows. All the results are repeated for 100 times with fixed initialization across all the cases and are consistent with **Theorem** 1. Thus, the convergence rate of the **TV-HMM** increases with the size of observations.

To illustrate the results of automatic relevance determination, we also visualize the $\pi_{k,i,j}$ before and after convergence, by taking the summation of $\{\Pi_k\}_{k=1}^{\tilde{K}}$. Results are shown on the right of Figure 2. The top right plot (c) shows the initial upper triangular transition matrix and the bottom plot (d) is the converged result from **Algorithm** 1. Note that the converged transition matrix is extremely sparse. Those non-zero spots on the diagonal indicate the existence of unnecessary regimes with size $0$. Other significant spots are near true change point locations, indicating the high relevance of these intervals with respect to the true regime. Then we infer the $\{Q(\mathbf{t}_k)\}_{k=1}^{\tilde{K}}$ for automatic model selection by leveraging the converged $\{\Pi_k\}_{k=1}^{\tilde{K}}$ as prior distributions.

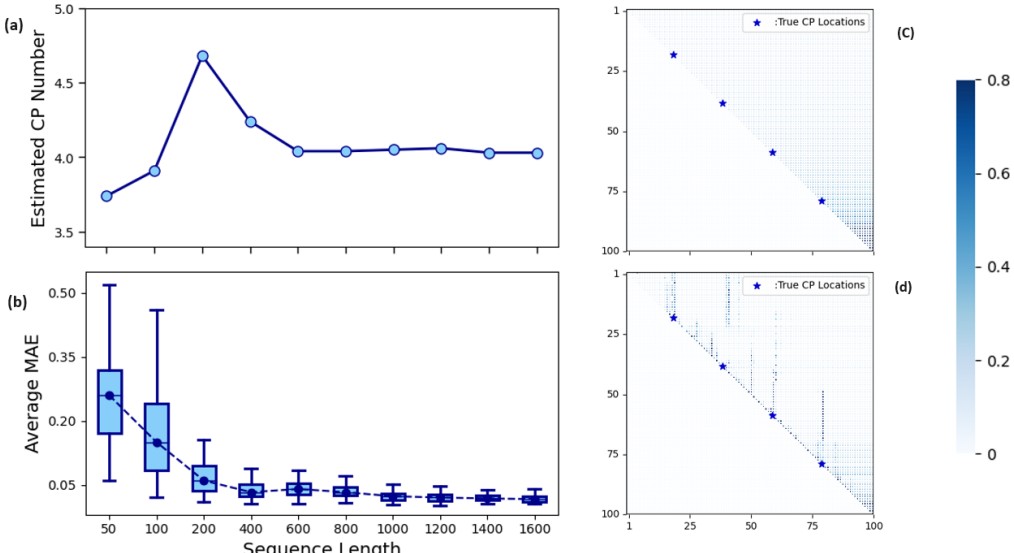

Figure 2: Left: The line plot (a) of the average estimated number of change points and the boxplot (b) of **MAE** varying with sequence length; Right: The heatmap of the sum of initial (c) and converged (d) $\tilde{K}$ transition matrices $[\Pi_1, ..., \Pi_{\tilde{K}}]$.

## 3.2 EVALUATION ON SIMULATED DATA

**Effectiveness** of our method is demonstrated by comparison with several well-developed CPD methods, including WBSLSW (Korkas & PryzlewiczV, 2017), ECP3O (Zhang et al., 2017), KCP (Harchaoui & Cappé, 2007), $\mathcal{D}_m$-BOCD (Altamirano et al., 2023) and another HMM-based method, DPHMM (Ko et al., 2015). The performance of CPD is measured by the Rand index, which is the similarity between two different data partitions (Lajugie et al., 2014; Fleming et al., 2023). It produces a value between 0 and 1, where 1 indicates perfect agreement.

|  | Model 1 | Model 2 | Model 3 |
|---|---|---|---|
| WBSLSW | 0.9068 | 0.3596 | 0.3849 |
| ECP3O | 0.9156 | 0.9580 | **0.9737** |
| DPHMM | **0.9637** | 0.8727 | 0.8869 |
| KCP | 0.9501 | 0.8436 | 0.8836 |
| $\mathcal{D}_m$-BOCD | 0.8123 | 0.8411 | 0.8413 |
| TV-HMM | 0.9523 | **0.9756** | 0.9615 |

Table 1: The performance of different CPD methods measured by the Rand index.

We consider three change-point models for the simulation, each with a significant characteristic. (Matteson & James, 2014; Chang et al., 2019). For Model 1, each regime follows either a binomial, Poisson, or normal distribution, with corresponding parameter variations. For Model 2, sequences are generated from 5-dimensional normal distributions, with either mean or covariance matrix shifts, and Model 3 increases the dimension to 10. Our simulations cover all common regime shifts in the piecewise i.i.d setting. For more details about the simulation setups, please refer to Appendix D.3.

Table 1 shows the performance of all methods for all cases. For Model 1, the accuracy of our methods is in line with DPHMM and outperforms the other four methods. Our method is the best among all candidate methods in Model 2. For Model 3, our method is also comparable to the best method ECP3O. The results indicate that our method performs consistently across all three models while existing methods suffer from fluctuation in performance.

|  | D=1 | D=5 | D=10 |
|---|---|---|---|
| MSE($\hat{u}$).Mean | 0.1885 | 0.1286 | 0.1868 |
| MSE($\hat{u}$).SD | $\pm$ 0.1635 | $\pm$ 0.0625 | $\pm$ 0.0971 |
| MSE($\hat{\Lambda}$).Mean | 0.9593 | 1.3382 | 4.1460 |
| MSE($\hat{\Lambda}$).SD | $\pm$ 1.7317 | $\pm$ 0.5381 | $\pm$ 0.1345 |

Table 2: MSE of the estimated posterior parameters $u$ and $\Lambda$

The proposed **TV-HMM** is able to simultaneously estimate the characteristics of each estimated regime, which is the mean and precision for **Equation** 1. We test the proposed method under different data dimensions. The parameter estimation is measured by the Mean Squared Error(MSE). We summarize the results in Table 2. Our method provides promising estimation results since the MSE of the estimated posterior mean ($\hat{u}$) and ground truth ($u$) falls within the range of 0.1 to 0.2 in all cases. For the posterior precision ($\hat{\Lambda}$) estimation, MSE is relatively larger than the other cases, which is reasonable since the number of parameters grows substantially with dimension $D$. Furthermore, the small standard deviation (SD) of MSE indicates the stability of our estimation across all setups.

### 3.3 Evaluation on real-world dataset

**Robustness** of our method is evaluated on the Well-log dataset from the real-world application. The data contains 4050 nuclear magnetic resonance measurements during the drilling procedures (Ruanaidh & Fitzgerald, 2012). Note that this sequence is corrupted by outliers, which have a significant effect on change point detection. To tackle this problem, Altamirano et al. (2023) develop the $\mathcal{D}_m-$BOCD that is incorporated with diffusion score matching, to reduce the effect of outliers on change point detection. This adaptation allows $\mathcal{D}_m$-BOCD to work on the corrupted dataset. Therefore, we compare the estimated locations of **TV-HMM** with their results, and the comparison is shown in Figure 3. The detected regime is separately colored, indicating the existence of a distributional shift. Most of the outliers are not identified as change points, and the results of **TV-HMM** are essentially in line with that in (Altamirano et al., 2023), which are plotted in a color bar at the bottom. The grey band indicates the mismatch of detected regimes. There is a clear change point at the time stamp 1540 that is not identifiable using $\mathcal{D}_m-$BOCD. Therefore, our method exhibits a comparative advantage on the Well-log dataset and demonstrates robustness to outliers.

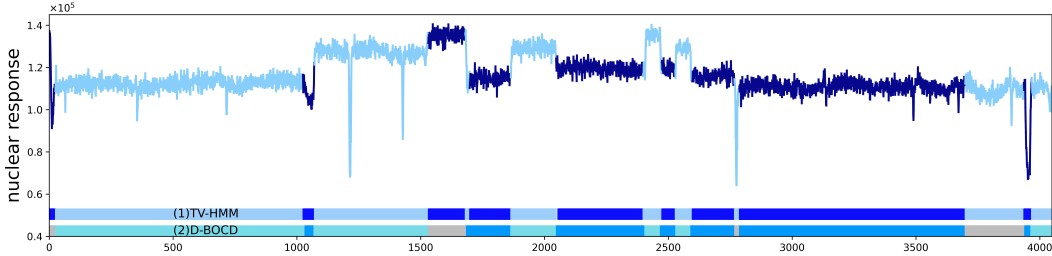

Figure 3: Estimated change point locations of Well-log data, color band (1) represents estimated regimes from **TV-HMM**, (2) represents estimated regimes from $\mathcal{D}_m-$BOCD. The grey bands represent the mismatches between the two methods.

## 4 Extension of **TV-HMM** with Maximum mean discrepancy

Previous results are developed based on the parametric likelihood function. Here, we alleviate the assumption using the kernel approach and propose a semi-supervised **TV-HMM** that is robust against outliers and model misspecification. Our motivation is that the expected log-likelihood term in the message function can be regarded as a distance measure between the observations subset $\mathbf{Y}_k$ and the characteristics of $k$-th regime $\zeta_k$. Thus, we can generalize the message functions of **Equation** 3 using Maximum mean discrepancy(MMD):

$$\mu_{\to \mathbf{t}_k}(n) = \sum_{m=1}^{n} \left\{ \mu_{\to \mathbf{t}_{i-1}}(m) \cdot \pi_{k,m,n} \cdot \exp\left[ -\frac{n-m+1}{G} \left\| \mathbb{E}_{\hat{P}_m^n} \varphi(y) - \mathbb{E}_{Q(\zeta_k)} \varphi(\zeta_k) \right\|_{\mathcal{H}} \right] \right\},$$

$$\mu_{\mathbf{t}_{k-1}\leftarrow}(m) = \sum_{n=m}^{N} \left\{ \mu_{\mathbf{t}_k \leftarrow}(n) \cdot \pi_{k,m,n} \cdot \exp\left[ -\frac{n-m+1}{G} \left\| \mathbb{E}_{\hat{P}_m^n} \varphi(y) - \mathbb{E}_{Q(\zeta_k)} \varphi(\zeta_k) \right\|_{\mathcal{H}} \right] \right\},$$

(4)

where $\hat{P}_m^n$ denotes the empirical distribution consisting of $n-m+1$ successive observations starting from time index $m$ to $n$, and $\varphi : \mathbb{R}^D \to \mathcal{H}$ represents the mapping to reproducing kernel Hilbert

---

**Algorithm 2** Training Procedure for Semi-Parametric Time-Varying Hidden Markov Model

---

**Input:** Observed sequence $\mathbf{Y}$; Initial number change points $\tilde{K}$; Maximum Iteration $I$; Step size $\eta$; Number of posterior samples $S$;

**Output:** Variational distributions $\{Q_\Phi(\zeta_k)\}_{k=1}^{\tilde{K}+1}$; Marginal probability of change point locations $\{Q(\mathbf{t}_k)\}_{k=1}^{\tilde{K}}$;

1: Initialization of $\{Q_\Phi(\zeta_k)\}_{k=1}^{K+1}$ with the distributions of initial regimes;
2: **for** $1 \leq i \leq I$ **do**
3:     **for** $1 \leq k \leq K+1$ **do**
4:         Sample $\{\zeta_k^s\}_S \sim Q_\Phi(\zeta_k)$ ; Compute $\|\mathbb{E}_{\hat{P}_m^n} \varphi(y) - \frac{1}{S} \sum_{s=1}^{S} \varphi(\zeta_k^s)\|_{\mathcal{H}}$ for any $1 \leq m \leq n \leq N$;
5:     **end for**
6:     Update $\{Q_\mathbf{t}^k(n,m)\}_{k=2}^{K}$ using message functions of Equation 4; $\pi_{k,m,n} \leftarrow \pi_{k,m,n} + \eta \cdot Q_\mathbf{t}^k(n,m)$
7:     Compute $\mathcal{J} \leftarrow$ MMD-ELBO using Equation 4; Update $\Phi \leftarrow \Phi + \eta \cdot \frac{\partial \mathcal{J}}{\partial \Phi}$
8: **end for**

---

space $\mathcal{H}$, and $G$ is a constant that adjusts the value of MMD. Unlike **Equation** 2 in the parametric model, where $Q(\theta)$ must be derived using variational inference, $Q(\zeta)$ can be generally modeled using non-parametric density estimation (Botev et al., 2010) and deep generative models (Kingma & Welling, 2013; Rezende et al., 2014)]. Denoting the distribution of $\zeta$ as $Q_\Phi(\zeta)$, where $\Phi$ is the model parameters, e.g. the weight values of neural networks, we propose a new MMD-based evidence lower bound (MMD-ELBO) as the objective function for $\Phi$ learning. The new loss function improves the robustness by replacing the likelihood functions in the original ELBO with a kernel-embedded distance. The formula of MMD-ELBO is given by:

$$\sum_{k=1}^{K+1} \sum_{m=1}^{N} \sum_{n \geq m}^{N} \frac{(m-n-1) \cdot Q_\mathbf{t}^k(n,m)}{G} \big\| \mathbb{E}_{\hat{P}_m^n} [\varphi(y)] - \mathbb{E}_{Q_\Phi(\zeta_k)} [\varphi(\zeta_k)] \big\|_{\mathcal{H}} + \mathrm{KL}(Q_\Phi(\zeta_k)\|p(\zeta_k)),$$

where $Q_\mathbf{t}^k(n,m)$ denotes the joint variational probability that $\mathbf{t}_k(n) = 1$ and $\mathbf{t}_{k-1}(m) = 1$ obtained from MMD-based message passing of **Equation** 4. For each iteration, we can evaluate the value of MMD-ELBO by sampling from $Q_\Phi(\zeta_k)$ and update $\Phi$ using the re-parameterization trick (Kingma et al., 2015). The pseudo-code of semi-parametric change point detection is summarized in **Algorithm** 2. We illustrate the performance of semi-parametric **TV-HMM** through three non-Gaussian examples, where the underlying sequence is generated from Poisson, chi-squared, and exponential distribution, respectively. The setup of the simulations can be found in Appendix D.4. Our performance is promising for all cases in terms of the Rand index, which is $0.9447$ for Poisson, $0.8686$ for chi-squared, and $0.8911$ for exponential distribution. Note that we do not incorporate any distributional knowledge as prior, the results indicate our method has robust performance over a broader class of data distributions.

**Relation with Parametric TV-HMM**: We illustrate its relation with the previously-discussed parametric **TV-HMM**. Under the Gaussian assumption with fixed variance, the likelihood $\mathbb{E}_{Q(\theta_k)} \ln p(\mathbf{Y}_k \mid \theta_k, \mathbf{t}_{k-1} = m, \mathbf{t}_k = n)$ in previous messenger passing **Equation** 3 is proportional to:

$$-\sum_{i=m}^{n} \mathbb{E}_{u_k}(y_i - u_k)^T \Lambda_k(y_i - u_k) \propto -(n-m+1) \cdot \| \sqrt{\Lambda_k}\mathbb{E}_{\hat{P}_m^n} [y] - \sqrt{\Lambda_k}\mathbb{E}_{Q(u_k)} [u_k] \|^2,$$

which is a special case of MMD with linear mapping $\varphi(x) = \sqrt{\Lambda_k}x$.

## 5 CONCLUSION

In this paper, we present **TV-HMM**, a time-varying Hidden Markov Model that enables simultaneous detection of change points and estimation of regime characteristics. Our method utilizes a variational EM algorithm incorporating stochastic approximation, and we prove its convergence rate for each change point location. Furthermore, we prove that our algorithm consistently selects the true number and locations of change points. Extensive numerical experiments provide evidence for our theoretical results and demonstrate the promising performance of our approach. In cases where the data distributions are unknown, we generalize our method using MMD and propose semi-parametric **TV-HMM** that does not rely on any distributional assumption. However, a limitation of current research is that CPD methods are primarily established on the piecewise i.i.d setting. In the future, we hope to extend our framework to a boarder class of CPD settings.

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

## A PRELIMINARY

### A.1 CHANGE POINT DETECTION METHODS

**WBSLSW**: The WBSLSW method (Korkas & PryzlewiczV, 2017) incorporates the non-parametric locally stationary wavelet process with wild binary segmentation, and can detect the second-order structure of the sequence with an unknown number of change points. We implement the WBSLSW using the R package `wbsts`.

**KCP**: The KCP (Harchaoui & Cappé, 2007) method is a dynamic programming method with a known number of change points. This algorithm detects the change points by minimizing the kernel least-squares criterion. In our cases, we combine KCP with a linear penalty pruning the number of change points. This is done by using the python package `ruptures` (Truong et al., 2020) with a Gaussian kernel and default parameters.

**DPHMM**: The DPHMM method developed by Ko et al. (2015) is a combination of the Dirichlet process and the Hidden Markov model to detect change points using MCMC. This method allows the number of change points to be unknown. We implement this algorithm using the R package `dirichletprocess` with default parameters and set $\tilde{K} = 10$

**ECP3O**: Zhang et al. (2017) propose a new change point search framework called change point procedure via pruned objectives. The ECP3O method uses the new search frame with the E-statistics which measures the goodness-of-fit. This method is implemented using `e.cp3o-delta` function with default parameters in R package `ecp`. (Nicholas A. James and Wenyu Zhang and David S. Matteson, 2019). The maximum number of change points $\tilde{K}$ is 10.

$\mathcal{D}_m$**-BOCD**: $\mathcal{D}_m$-BOCD is an online method developed by Altamirano et al. (2023). This method is generalized from BOCD (Adams & MacKay, 2007) with diffusion score matching, which is robust to sequences with outliers. We implement this method using the code provided on the author's GitHub page.

### A.2 VARIATIONAL INFERENCE

Variational inference (VI) Blei et al. (2017) works as a fast approximation method for Bayesian inference. Given the observation $\mathbf{x}$ and latent variable $\mathbf{z}$, VI uses a tractable variational distribution $q$ drawn from the function class $\mathcal{F}$ to approach the complicated posterior $p(\mathbf{z} \mid \mathbf{x})$ by minimizing their KL divergence. However, the KL can not be computed analytically. Classical VI optimizes an alternative objective called Evidence Lower Bound (ELBO) that is equivalent to log marginal likelihood minus the KL:

$$
\begin{aligned}
\ln p(\mathbf{x}) &= \text{ELBO}(q) + \text{KL}(q \| p(\mathbf{z} \mid \mathbf{x})) \\
&= \int q(\mathbf{z}) \ln \left\{ \frac{p(\mathbf{x}, \mathbf{z})}{q(\mathbf{z})} \right\} d\mathbf{z} - \int q(\mathbf{z}) \ln \left\{ \frac{p(\mathbf{z} \mid \mathbf{x})}{q(\mathbf{z})} \right\} d\mathbf{z}.
\end{aligned}
$$

Under the common mean-field assumption $q(\mathbf{z}) = \prod_{j=1}^{m} q_j(z_j)$, the maximizer of ELBO $q_j^*$ has the analytical solution $q_j^*(z_j) \propto \exp \left\{ \mathbb{E}_{\mathbf{z}_{-j}} \left[ \log p(\mathbf{z}, \mathbf{x}) \right] \right\}$, where $\mathbf{z}_{-j}$ denotes all variables $z_i$ other than $z_j$, that can be solved by the coordinate ascent algorithm.

Recently, VI has been commonly applied in training deep generative models, including VAE (Kingma & Welling, 2013) and deep diffusion model (Ho et al., 2020) to approximate complex posterior distributions. VI plays a crucial role in approximating the posterior distribution over the latent variables, enabling efficient learning and generation of high-quality samples from complex data distributions.

## B NORMAL MEAN-VARIANCE SHIFT MODEL

In this section, we derive updating formulas for the Normal Mean-Variance Shift model. Denoting the set of all latent variables as $\boldsymbol{\xi} = \{\{\mathbf{t}_k\}_{k=1}^{K}, \{\theta_{K+1}\}_{k=1}^{K+1}\}$ and the hyperparameters set $\alpha_k =$

$\{\beta, \nu_0, V_0\}$, we assume a constrained mean-field $Q$ family in variational inference:

$$Q(\boldsymbol{\xi}) = Q(\mathbf{t}_1) \prod_{k=2}^{K} Q\left(\mathbf{t}_k | \mathbf{t}_{k-1}\right) \prod_{k}^{K+1} Q\left(u_k\right) Q(\Lambda_k).$$

Then the variational lower bound is given by

$$
\begin{aligned}
\mathcal{L}(Q) = {} & \sum_{k-1}^{K+1} \left[ \sum_{t_k, t_{k-1}} Q\left(\mathbf{t}_k, \mathbf{t}_{k-1}\right) \int Q\left(u_k\right) Q(\Lambda_k) \ln p\left(\mathbf{Y}_k \mid \mathbf{t}_k, \mathbf{t}_{k-1}, u_k, \Lambda_k\right) du_k d\Lambda_k \right] \\
& + \sum_{k=1}^{K+1} \left[ \int Q\left(u_k\right) Q(\Lambda_k) \ln \frac{\mathcal{N}\left(u_k; 0, \beta^{-1} I\right) \mathcal{W}\left(\Lambda_k; \nu^0, V^0\right)}{Q\left(u_k\right) Q_T(\Lambda_k)} du_k d\Lambda_k \right] \\
& + \sum_{k=1}^{K} \left[ \sum_{t_k, t_{k-1}} Q_t\left(\mathbf{t}_k, \mathbf{t}_{k-1}\right) \ln \frac{p\left(\mathbf{t}_k \mid \mathbf{t}_{k-1}\right)}{Q_t\left(\mathbf{t}_k \mid \mathbf{t}_{k-1}\right)} \right].
\end{aligned}
$$

Minimizing KL divergence leads to an analytical solution. We can directly apply it to give the optimal solutions for the family of factors $Q$ of variational posteriors

$$Q(\mathbf{t}_1) = \prod_{i=1}^{N} \tilde{\pi}_{1,i}^{\mathbf{t}_1(i)}, \quad Q\left(\mathbf{t}_k | \mathbf{t}_{k-1}\right) = \prod_{i=1}^{N} \prod_{j=1}^{N} \hat{\pi}_{k,i,j}^{\mathbf{t}_k(i) \times \mathbf{t}_{k-1}(j)},$$

$$Q(u_k) = \mathcal{N}\left(u_k \mid m_k, L_k^{-1}\right), \quad Q(\Lambda_k) = \mathcal{W}\left(\Lambda_k \mid \nu_k, V_k\right).$$

Given prior distributions defined above, solutions for variational parameters are given by

$$
\begin{aligned}
m_k = {} & \left[ \langle \Lambda_k \rangle \sum_{n=1}^{N} \sum_{m \geq n}^{N} Q\left(\mathbf{t}_k(m), \mathbf{t}_{k-1}(n)\right) \sum_{j=n}^{m} \mathbf{1} + \mathbf{I}/\beta \right]^{-1} \\
& \times \left[ \langle \Lambda_k \rangle \sum_{n=1}^{N} \sum_{m \geq n}^{N} Q\left(\mathbf{t}_k(m), \mathbf{t}_{k-1}(n)\right) \sum_{j=n}^{m} y_j \right], \\
L_k = {} & \langle \Lambda_k \rangle \sum_{n=1}^{N} \sum_{m \geq n}^{N} Q\left(\mathbf{t}_i(m), \mathbf{t}_{i-1}(n)\right) \sum_{j=n}^{m} \mathbf{1} + \alpha^{-1} \mathbf{I}, \\
\nu_i = {} & \nu_0 + \sum_{n=1}^{N} \sum_{m \geq n}^{N} Q\left(\mathbf{t}_i(m), \mathbf{t}_{i-1}(n)\right) \sum_{j=n}^{m} \mathbf{1},
\end{aligned}
$$

and

$$V_k^{-1} = V_0^{-1} + \sum_{n=1}^{N} \sum_{m \geq n}^{N} Q\left(\mathbf{t}_k(m), \mathbf{t}_{k-1}(n)\right) \left[ \sum_{j=n}^{m} y_j y_j^\top - 2 \sum_{j=n}^{m} y_j \langle u_k \rangle^\top + \sum_{j=n}^{m} \langle u_k u_k^\top \rangle \right].$$

As we mentioned in **Section** 2.2, solutions for $Q_t\left(\mathbf{t}_k \mid \mathbf{t}_{k-1}\right)$ can be obtained through sum-product algorithm. The updating formulas are given by

$$Q\left(\mathbf{t}_k(n) = 1\right) = \mu_{\rightarrow \mathbf{t}_k}(n) \cdot \mu_{\mathbf{t}_k \leftarrow}(n) = \tilde{\pi}_{k,n},$$

and

$$
\begin{aligned}
& Q\left(\mathbf{t}_{k-1}(m) = 1, \mathbf{t}_k(n) = 1\right) \\
= {} & \mu_{\rightarrow \mathbf{t}_{k-1}}(m) \cdot \pi_{i,m,n} \cdot \exp\left(\mathrm{E}_{Q(u_k) Q(\Lambda_k)} \ln p(\mathbf{Y}_k, \mathbf{t}_k, u_k, \Lambda_k \mid \mathbf{t}_{k-1})\right) \cdot \mu_{t_k \leftarrow}(n),
\end{aligned}
$$

where messages are obtained recursively. For $k = 1, \ldots, K$,

$$
\begin{aligned}
\mu_{\rightarrow \mathbf{t}_k}(n) = {} & \sum_{m=1}^{n} \left\{ \mu_{\rightarrow \mathbf{t}_{k-1}}(m) \cdot \pi_{k,m,n} \cdot \exp\left\{ \sum_{j=m}^{n} \frac{1}{2} \langle \ln |\Lambda_k| \rangle \right. \right. \\
& \left. \left. - \frac{1}{2} \sum_{j=m}^{n} \mathrm{Tr}\left[ \left(y_j \cdot y_j^\top - y_j \cdot \langle u_k^\top \rangle - \langle u_k \rangle \cdot y_j^\top + \langle u_k, u_k^\top \rangle\right) \cdot \langle \Lambda_k \rangle \right] \right\} \right\},
\end{aligned}
$$

and

$$
\begin{aligned}
\mu_{\mathbf{t}_{k-1}\leftarrow}(m) \quad = \quad & \sum_{n=m}^{N} \left\{ \mu_{\mathbf{t}_{k-1}\leftarrow}(n) \cdot \pi_{k,m,n} \cdot \exp\left\{ \sum_{j=m}^{n} \frac{1}{2} \left\langle \ln|\Lambda_k| \right\rangle \right. \right. \\
& \left. \left. - \frac{1}{2} \sum_{j=m}^{n} \operatorname{Tr}\left[ \left( y_j \cdot y_j^\top - y_j \cdot \langle u_k^\top \rangle + \langle u_k, u_k^\top \rangle \right) \cdot \langle \Lambda_k \rangle \right] \right\} \right\}.
\end{aligned}
$$

To start recursion, the initial message state $\mu_{\to \mathbf{t}_1}$ and $\mu_{\mathbf{t}_K\leftarrow}$ are given by

$$
\begin{aligned}
\mu_{\to \mathbf{t}_1}(m) \quad = \quad & \pi_{1,m} \exp\left\{ \sum_{j=m}^{n} \frac{1}{2} \left\langle \ln|\Lambda_1| \right\rangle \right. \\
& \left. - \frac{1}{2} \sum_{j=m}^{n} \operatorname{Tr}\left[ \left( y_j \cdot y_j^\top - y_j \cdot \langle u_1^\top \rangle - \langle u_1 \rangle \cdot y_j^\top + \langle u_1, u_1^\top \rangle \right) \cdot \langle \Lambda_1 \rangle \right] \right\},
\end{aligned}
$$

and

$$
\begin{aligned}
\mu_{\mathbf{t}_K\leftarrow}(m) \quad = \quad & \exp\left\{ \sum_{j=m}^{n} \frac{1}{2} \left\langle \ln|\Lambda_{K+1}| \right\rangle \right. \\
& \left. - \frac{1}{2} \sum_{j=m}^{n} \operatorname{Tr}\left[ \left( y_j \cdot y_j^\top - y_j \cdot \langle u_{K+1}^\top \rangle + \langle u_{K+1}, u_{K+1}^\top \rangle \right) \cdot \langle \Lambda_{K+1} \rangle \right] \right\},
\end{aligned}
$$

where we have assumed:

$$
\langle u_k \rangle = m_k, \quad \langle u_k u_k^\top \rangle = m_k m_k^\top + L_k^{-1}, \quad \langle \Lambda_k \rangle = \nu_k V_k,
$$

$$
\langle \ln|\Lambda_k| \rangle = \sum_{j=1}^{D} \psi\left( \frac{u_k + 1 - j}{2} \right) + D\ln 2 + \ln|V_k|,
$$

and $\psi(\cdot)$ is the digamma function.

## C  PROOF OF THEOREM 1

To start up, it's worth mentioning that practically for a sequence of time $T$, we observe finite data points $\{y_1, ..., y_T\}$ at each time stamp $t = 1, ..., T$, which is the input for the proposed algorithm. However, in theory, we consider a continuous timeline and there are infinitely many data points between any time intervals $[m, n] \subseteq [0, T]$. Thus before discussing our theoretical results, we first list our setup and assumptions:

**A1**: The underlying sequence on time interval $[0, T]$ consists of $K$ change points $0 < T_1 < ... < T_K < T$ with $T_0 = 0$ and $T_{K+1} = T$. For any time stamp $T_{k-1} < t < T_k$, the random function $y(t) : \mathbb{R} \to \mathbb{R}^D$ represents the sample drawn from $\mathcal{N}(y \mid u_k, \Lambda_k)$ at time $t$.

**A2**: The total number of collected observations is $N$. For any time interval $[m, n] \subseteq [0, T]$, the number of observations within this interval equals $O(N^{\frac{n-m}{T}})$.

**A3**: The algorithm initializes $M_{K+1} > K + 1$ regimes corresponding to $\{\mathbf{t}_i\}_{i=1}^{M_{K+1}-1}$ change points. The regimes are segmented by a time subset $\{t_1, ..., t_{M_{K+1}-1}\}$ with equidistance, such that $t_{i+1} - t_i = \frac{T}{M_{K+1}}$. Based on the characteristic of the regime between $[t_i, t_{i+1}]$, we can further categorize $\{\mathbf{t}_i\}_{i=1}^{M_{K+1}-1}$ into two subsets:

- Any $\mathbf{t}_i \in \{\mathbf{t}_{M_k}\}_{k=1}^{K}$ denotes the junction points, e.g In initialization, there is a true change point $T_k$ located within the interval $[t_{i-1}, t_i]$ and $y(t)$ for $t \in [t_{i-1}, t_i]$ does not identically distributes.
- For $k = 1, ..., K+1$, any $i \in \{M_{k-1} + 1, ..., M_k - 1\}$ denotes the non-junction index and we have $T_{k-1} < t_i < T_k$, where we let $M_0 = 0$. Every $y(t)$ for $t \in [t_{i-1}, t_i]$ distributes equivalently with those in $[T_k, T_{k+1}]$.

**A4**: The row of transition matrix $\Pi_k$ is a uniform distribution, such that $\pi_{k,i,j} = N^{\frac{iT}{T}}$. The observation dimension $D$, the number of change point $K$ and initialized change point number $M_{K+1} - 1$ is fixed.

We further define the random functions of $a(t)$, $b(t)$ and $c(t)$ for time interval $[m, n]$ as following:

$$\int_m^n a(t)dt = \begin{cases} \int_0^{m-n} -b(t)dt & \text{if } n \leq m, \\ \int_0^{n-m} c(t)dt & \text{if } n > m. \end{cases}$$

with

$$b(t) = \max_k \left[ \ln |\Lambda_k| / 2 - (y(t) - u_k)^\top \Lambda_k (y(t) - u_k) / 2 \mid t \in [T_{k-1}, T_k] \right],$$

$$c(t) = \max_k \left[ \ln |\Lambda_k| / 2 - (y(t) - u_k)^\top \Lambda_k (y(t) - u_k) / 2 \mid t \notin [T_{k-1}, T_k] \right].$$

Intuitively, the defined $b(t)$ is the maximum likelihood value at time $t$, where the likelihood function is parameterized with true $u$ and $S$, while $c(t)$ is the maximum likelihood value associated with false parameters $u$ and $S$. Thus, the integral range $[m, n]$ of $b(t)$ and $c(t)$ indicates the sequence length that is correctly aligned or not, respectively.

**Corollary 1.** *As $N$ approaches infinity, for any time interval $[m, n]$, the random variables, we have:*

$$\int_m^n (c(t) - b(t)) \, dt = \mathcal{O}_p(N^{\frac{n-m}{T}}) < 0.$$

*Proof: First using the Lemma from (Bishop et al., 2007):*

**Lemma 1.** *Let $\{X_n\}$ be a stochastic sequence with $\mu_n = \mathbb{E}(X_n)$ and $\sigma_n^2 = \text{Var}(X_n) < \infty$, then $X_n = \mu_n + O_p(\sigma_n)$.*

*Thus, based on the **Lemma 1** and our assumptions, it's easy to see the following results hold:*

*1): The value of $\int_m^n b(t)dt$ equals to:*

$$N^{\frac{n-m}{T}} \max_k \left( \frac{1}{2} \ln |\Lambda_k| - \frac{1}{2} \text{Tr} \left( \left[ \mathbb{E}\left[ y(t)y(t)^\top \right] - \mathbb{E}[y(t)]u_k^\top - u_k \mathbb{E}[y(t)^T] + u_k u_k^T \right] \cdot \Lambda_k \right) \right)$$

$$= N^{\frac{n-m}{T}} \max_k \left( \frac{1}{2} \ln |\Lambda_k| \right) + \mathcal{O}_p(N^{\frac{n-m}{2T}}).$$

*2): The value of $\int_m^n c(t)dt$ eqauls to:*

$$N^{\frac{n-m}{T}} \max_k \max_{k' \neq k} \left( \frac{1}{2} \ln |\Lambda_k| - \frac{1}{2} \left[ (\mu_k - \mu_{k'})^\top \Lambda_k (\mu_k - \mu_{k'}) + \text{Tr}(\Lambda_{k'}^{-1} \cdot \Lambda_k) \right] \right) + \mathcal{O}_p(N^{\frac{n-m}{2T}}).$$

Then the value of $\int_m^n (c(t) - b(t)) \, dt$ is given by:

$$\int_m^n (c(t) - b(t)) \, dt$$

$$= N^{\frac{n-m}{T}} \max_k \max_{k' \neq k} - \frac{1}{2} \left[ (\mu_k - \mu_{k'})^\top \Lambda_k (\mu_k - \mu_{k'}) + \text{Tr}(\Lambda_{k'}^{-1} \cdot \Lambda_k) \right] + \mathcal{O}_p(N^{\frac{n-m}{2T}}) < 0.$$

$\square$

Based on the above assumptions, we can present our results in the following:

**Theorem 2.** *For $t_i \in \{t_{M_{k-1}+1}, ..., t_{M_k}\}$, the value of each forward message is given by:*

$$\mu_{\to \mathbf{t}_i}(n) = \frac{N^{\frac{(i-1)n}{T}}}{N^i (\ln N)^{i-1}} \exp\left( \int_0^{T_k} b(t)dt \right) \exp\left( \int_0^{n-T_k} \alpha(t)dt \right).$$

**Proof:** We will use the method of induction to drive the general formula of the forward message. By considering each data $y(t)$ as the continuous function of time $t$, the initial message is given by:

$$
\begin{aligned}
\mu_{\to \mathbf{t}_1}(m) &= \pi_{1,m} \cdot \exp\left\{ \int_0^m \left\{ \frac{1}{2} \left\langle \ln|\Lambda_1^0| \right\rangle \right.\right. \\
&\quad \left.\left. - \frac{1}{2} \operatorname{Tr}\left[ \left( y(t) \cdot y(t)^\top - y(t) \cdot \langle u_1^0 \rangle^\top - \langle u_1^0 \rangle \cdot y(t)^\top + \langle u_1^0, u_1^{0\top} \rangle \right) \cdot \langle \Lambda_1^0 \rangle \right] \right\} \right\} \\
&= \begin{cases} \mathcal{O}_p(\pi_{1,m} \cdot \exp\{\int_0^m b(t)dt\}) & \text{if } m \le T_1, \\[2mm] \mathcal{O}_p(\pi_{1,m} \cdot \exp\{\int_0^{T_1} b(t)dt + \int_{T_1}^m c(t)dt\}) & \text{if } m \ge T_1. \end{cases} \\
&= \mathcal{O}_p\left( \frac{1}{N} \exp\left( \int_0^{T_1} b(t)dt \right) \cdot \exp\left( \int_{T_1}^m \alpha(t)dt \right) \right),
\end{aligned}
$$

where we use the fact that $\pi_{i,m} = 1/N$ and the initial segment is a subset of the first regime $[0, t_1] \subset [0, T_1]$ and the initialized parameters are consistent with the true value $S_1$ and $u_1$, such that:

$$
\langle u_1^0 \rangle = \hat{u}_1 \xrightarrow{\text{P}} u_1, \quad \langle u_1^0, u_1^{0\top} \rangle = \hat{u}_1 \cdot \hat{u}_1^\top, \quad \langle \Lambda_1^0 \rangle = \hat{S}_1 \xrightarrow{\text{P}} S_1, \quad \langle \ln|\Lambda_1^0| \rangle = \ln\left|\hat{S}_1\right| \xrightarrow{\text{P}} \ln|S_1|.
$$

Now consider the next message using the updated formula:

$$
\begin{aligned}
\mu_{\to \mathbf{t}_2}(n) &= \int_0^n \left\{ \mu_{f_0 \to \mathbf{t}_1}(m) \cdot \pi_{2,m,n} \cdot \exp\left\{ \int_m^n \frac{1}{2} \left\langle \ln|\Lambda_2^0| \right\rangle \right.\right. \\
&\quad \left.\left. - \frac{1}{2} \operatorname{Tr}\left[ \left( y(t) \cdot y(t)^\top - y(t) \cdot \langle u_2^0 \rangle^\top - \langle u_2^0 \rangle y(t)^\top + \langle u_2^0, u_2^{0\top} \rangle \right) \cdot \langle \Lambda_2^0 \rangle \right] dt \right\} \right\} dm \\
&= \begin{cases} \mathcal{O}_p\left( \frac{1}{N}(\int_0^n N^{\frac{m-T}{T}} dm) \cdot \exp\{\int_0^n b(t)dt\} \right) & \text{if } n \le T_1, \\[2mm] \mathcal{O}_p\left( \frac{1}{N}(\int_0^n N^{\frac{m-T}{T}} dm) \cdot \exp\{\int_0^{T_1} b(t)dt + \int_{T_1}^n c(t)dt\} \right) & \text{if } n \ge T_1. \end{cases} \\
&= \mathcal{O}_p\left( \frac{N^{\frac{n}{T}}}{N^2 \cdot \ln N} \exp\left( \int_0^{T_1} b(t)dt \right) \cdot \exp\left( \int_{T_1}^n \alpha(t)dt \right) \right).
\end{aligned}
$$

Therefore, the exponential term is exactly the same as the initial message. It's easy to see as long as $i \in \{1, ..., M_1 - 1\}$, the message is given by:

$$
\mu_{\to \mathbf{t}_i}(n) = \frac{N^{\frac{(i-1)n}{T}}}{N^i (\ln N)^{i-1}} \exp\left( \int_0^{T_k} b(t)dt \right) \exp\left( \int_0^{n-T_k} \alpha(t)dt \right).
$$

Now consider the first junction point $i = M_1$, the message is given by:

$$
\begin{aligned}
\mu_{\to \mathbf{t}_{M_1}}(n) &= \mathcal{O}_p\left( \int_0^n \left\{ \frac{N^{\frac{(M_1-1)m}{T}}}{N^{M_1} \cdot (\ln N)^{(M_1-2)}} \exp\left( \int_0^{T_1} b(t)dt \right) \right.\right. \\
&\quad \left.\left. \times \exp\left( \int_{T_1}^m \alpha(t)dt \right) \cdot \exp\left( \int_0^{n-m} c(t)dt \right) \right\} dm \right).
\end{aligned}
$$

We can discuss in part:

**1.** for $n \ge m \ge T_1$:

$$
\mu_{\to \mathbf{t}_{M_1}}(n) = \mathcal{O}_p\left( \frac{N^{\frac{(M_1-1)n}{T}}}{N^{M_1}(\ln N)^{(M_1-1)}} \exp\left( \int_0^{T_1} b(t)dt \right) \exp\left( \int_0^{n-T_1} c(t)dt \right) \right).
$$

**2**. for $n \geq T_1 \geq m$:

$$
\begin{aligned}
\mu_{\to \mathbf{t}_{M_1}}(n) &= \mathcal{O}_p\bigg( \frac{N^{\frac{(M_1-1)n}{T}}}{N^{M_1}(\ln N)^{(M_1-1)}} \int_0^n \exp\Big( \int_0^{T_1} b(t)dt \Big) \\
&\quad \times \exp\Big( \int_0^{T_1-m} (c(t)-b(t))dt \Big) \exp\Big( \int_0^{n-T_1} c(t)dt \Big) dm \bigg).
\end{aligned}
$$

**3**. for $T_1 \geq n \geq m$:

$$
\begin{aligned}
\mu_{\to \mathbf{t}_{M_1}}(n) &= \mathcal{O}_p\bigg( \frac{N^{\frac{(M_1-1)n}{T}}}{N^{M_1}(\ln N)^{(M_1-1)}} \int_0^n \exp\Big( \int_0^{T_1} b(t)dt \Big) \\
&\quad \times \exp\Big( \int_0^{n-m} (c(t)-b(t))dt \Big) \exp\Big( \int_0^{T_1-n} -b(t)dt \Big) dm \bigg).
\end{aligned}
$$

Thus,

$$
\mu_{\to \mathbf{t}_{M_1}}(n) = \mathcal{O}_p\left( \frac{N^{\frac{(M_1-1)n}{T}}}{N^{M_1}(\ln N)^{(M_1-1)}} \exp\Big( \int_0^{T_1} b(t)dt \Big) \exp\Big( \int_{T_1}^n \alpha(t)dt \Big) \right).
$$

Then we evaluate the first non-junction point $i = M_1 + 1$. By discussing it by part, we show that:

**1**. if $n \geq T_1$:

$$
\begin{aligned}
\mu_{\to \mathbf{t}_{M_1+1}}(n) &= \mathcal{O}_p\bigg( \exp\Big( \int_0^{T_2} b(t)dt \Big) \exp\Big( \int_{T_2}^n \alpha(t)dt \Big) \\
&\quad \times \int_0^n \left\{ \frac{N^{\frac{M_1 m}{T}}}{N^{M_1+1}(\ln N)^{(M_1-1)}} \exp\Big( \int^{T_1-m} (c(t)-b(t))dt \Big) \right\} dm \bigg).
\end{aligned}
$$

**2**. if $n \leq T_1$:

$$
\begin{aligned}
\mu_{\to \mathbf{t}_{M_1+1}}(n) &= \mathcal{O}_p\bigg( \exp\Big( \int_0^{T_2} b(t)dt \Big) \exp\Big( \int_{T_2}^n \alpha(t)dt \Big) \\
&\quad \times \int_0^n \left\{ \frac{N^{\frac{M_1 m}{T}}}{N^{M_1+1}(\ln N)^{(M_1-1)}} \exp\Big( \int^{n-m} (c(t)-b(t))dt \Big) \right\} dm \bigg).
\end{aligned}
$$

In both cases, we can rewrite the message as:

$$
\mu_{\to \mathbf{t}_{M_1+1}}(n) = \mathcal{O}_p\left( \frac{N^{\frac{M_1 m}{T}}}{N^{M_1+1} \cdot (\ln N)^{M_1}} \exp\Big( \int_0^{T_2} b(t)dt \Big) \exp\Big( \int_{T_2}^n \alpha(t)dt \Big) \right),
$$

which returns to the initial message $\mu_{\to \mathbf{t}_1}(n)$ with the same exponential terms. $\square$

By the same induction procedure, it's easy to see **Theorem** 2 holds for all $i \in \{t_{M_{k-1}+1}, ..., t_{M_k}\}$.

**Theorem 3.** *For $t_i \in \{t_{M_{k-1}}, ..., t_{M_k-1}\}$, the value of each backward message is given by:*

$$
\mu_{\mathbf{t}_i \leftarrow}(m) = \exp\left( \int_0^{T-T_{k-1}} b(t)dt \right) \exp\left( \int_0^{T_{k-1}-m} \alpha(t)dt \right).
$$

**Proof:** The proof of **Theorem** 3 is similar to that of **Theorem** 2. The initial backward message is given by:

$$
\mu_{\mathbf{t}_{M_{K+1}-1} \leftarrow}(m) = \exp\left\{ \int_m^T \left\{ \frac{1}{2} \left\langle \ln |\Lambda_{K+1}^0| \right\rangle - \frac{1}{2} \operatorname{Tr}\left[ \left( y(t)y(t)^\top - y(t) \cdot \left\langle u_{K+1}^0 \right\rangle^\top \right. \right. \right. \right.
$$

$$
\left. \left. \left. \left. - \left\langle u_{K+1}^0 \right\rangle^\top \cdot y(t) + \left\langle u_{K+1}^0, u_{K+1}^{0\top} \right\rangle \right) \cdot \left\langle \Lambda_{K+1}^0 \right\rangle \right] dt \right\} \right\}
$$

$$
= \begin{cases} \mathcal{O}_p(\exp\left\{ \int_m^T b(t)dt \right\}) & \text{if } m \geq T_K, \\[2mm] \mathcal{O}_p(\exp\left\{ \int_{T_K}^T b(t)dt + \int_m^{T_K} c(t)dt \right\}) & \text{if } m \leq T_K. \end{cases}
$$

$$
= \mathcal{O}_p\left( \exp\left( \int_0^{T-T_K} b(t)dt \right) \exp\left( \int_0^{T_K-m} \alpha(t)dt \right) \right),
$$

where the initialized parameters are consistent estimators of true $u_{K+1}$ and $S_{K+1}$. Now consider the next backward message using the updated formula:

$$
\mu_{\rightarrow \mathbf{t}_{M_{K+1}-2}}(m) = \int_m^T \left\{ \mu_{\rightarrow \mathbf{t}_{M_{K+1}-1}}(n) \cdot \pi_{M_{K+1}-1,m,n} \right.
$$

$$
\times \exp\left\{ \int_m^n \left\{ \frac{1}{2} \left\langle \ln |\Lambda_{K+1}^0| \right\rangle - \frac{1}{2} \operatorname{Tr}\left[ \left( y(t) \cdot y(t)^\top - 2y(t) \cdot \left\langle u_{K+1}^0 \right\rangle \right. \right. \right. \right.
$$

$$
\left. \left. \left. \left. + \left\langle u_{K+1}^0, u_{K+1}^{0\top} \right\rangle \right) \cdot \left\langle \Lambda_{K+1}^0 \right\rangle \right] dt \right\} \right\} dn
$$

$$
= \begin{cases} \mathcal{O}_p\left( N^{\frac{m-T}{T}} \cdot \exp\left\{ \int_0^{T-m} b(t)dt \right\} \right) & \text{if } m \geq T_K, \\[2mm] \mathcal{O}_p\left( N^{\frac{m-T}{T}} \cdot \exp\left\{ \int_0^{T-T_K} b(t)dt + \int_0^{T_K-m} c(t)dt \right\} \right) & \text{if } m \leq T_K. \end{cases}
$$

$$
= \mathcal{O}_p\left( \exp\left( \int_0^{T-T_K} b(t)dt \right) \exp\left( \int_0^{T_K-m} \alpha(t)dt \right) \right).
$$

Thus, it's easy to show for all $i \in \{M_K + 1, ..., M_{K+1} - 2\}$,

$$
\mu_{\mathbf{t}_i \leftarrow}(m) = \mathcal{O}_p\left( \exp\left( \int_0^{T-T_K} b(t)dt \right) \exp\left( \int_0^{T_K-m} \alpha(t)dt \right) \right).
$$

Then we consider the first junction point $i = M_K$:

$$
\mu_{\mathbf{t}_{M_K} \leftarrow}(m) = N^{\frac{m-T}{T}} \int_m^T \left\{ \exp\left( \int_0^{T-T_K} b(t)dt \right) \right.
$$

$$
\left. \times \exp\left( \int_0^{T_K-n} \alpha(t)dt \right) \exp\left( \int_0^{n-m} c(t)dt \right) \right\} dn.
$$

Consider three cases:

**1.** If $n \geq m \geq T_k$:

$$
\mu_{\mathbf{t}_{M_K} \leftarrow}(m) = N^{\frac{m-T}{T}} \int_m^T \left\{ \exp\left( \int_0^{T-T_K} b(t)dt \right) \right.
$$

$$
\left. \times \exp\left( \int_0^{m-T_K} -b(t)dt \right) \exp\left( \int_0^{n-m} (c(t) - b(t))dt \right) \right\} dn.
$$

**2**. If $n \geq T_K \geq m$:

$$
\mu_{\mathbf{t}_{M_K} \leftarrow}(m) = N^{\frac{m-T}{T}} \int_m^T \left\{ \exp\left(\int_0^{T-T_K} b(t)dt\right) \right.
$$
$$
\left. \times \exp\left(\int_0^{T_K-m} c(t)dt\right) \exp\left(\int_0^{n-T_K} (c(t)-b(t))dt\right) \right\} dn.
$$

**3**. If $T_K \geq n \geq m$:

$$
\mu_{\mathbf{t}_{M_K} \leftarrow}(m) = N^{\frac{m-T}{T}} \int_m^T \left\{ \exp\left(\int_0^{T-T_K} b(t)dt\right) \cdot \exp\left(\int_0^{T_K-m} c(t)dt\right) \right\} dn.
$$

Thus we can sum it up as:

$$
\mu_{\mathbf{t}_i \leftarrow}(m) = \mathcal{O}_p\left(\exp\left(\int_0^{T-T_K} b(t)dt\right) \exp\left(\int_0^{T_K-m} \alpha(t)dt\right)\right).
$$

When it comes to the new point in the previous segment $i = M_K - 1$:

**1**. If $n \leq T_K$:

$$
\mu_{\mathbf{t}_{M_K-1} \leftarrow}(m) = N^{\frac{m-T}{T}} \int_m^T \left\{ \exp\left(\int_0^{T-T_{K-1}} b(t)dt\right) \right.
$$
$$
\left. \times \exp\left(\int_0^{T_K-n} c(t)-b(t)dt\right) \exp\left(\int_0^{T_{K-1}-m} \alpha(t)dt\right) \right\} dn.
$$

**2**. If $n \geq T_K$:

$$
\mu_{\mathbf{t}_{M_K-1} \leftarrow}(m) = N^{\frac{m-T}{T}} \exp\left(\int_0^{T-T_{K-1}} b(t)dt\right)
$$
$$
\times \int_m^T \left\{ \exp\left(\int_0^{\min\{n-T_K, n-m\}} (c(t)-b(t))dt\right) \right.
$$
$$
\left. \times \exp\left(\int_0^{T_{K-1}-m} \alpha(t)dt\right) \right\} dn.
$$

Thus, following the same procedure in the proof of **Theorem** 2, we can derive that for all $i$, the recursive formula holds.

We are now ready to prove the location consistency. First consider the change point $t_i \in \{t_{M_{k-1}+1}, ..., t_{M_k-1}\}$. The unnormalized marginal probability $\tilde{Q}(t_i = m)$ is given by:

$$
\frac{N^{\frac{in}{T}}}{N^i(\ln N)^{i-1}} \exp\left(\int_0^{T+T_k-T_{k-1}} b(t)dt\right) \exp\left(\int_0^{m-T_k} \alpha(t)dt\right) \exp\left(\int_0^{T_{k-1}-m} \alpha(t)dt\right).
$$

Thus we can discuss all possible values of location $m$:

**1**. When $T_{k-1} \leq m \leq T_k$: It's easy to show

$$
\int_0^{m-T_k} \alpha(t)dt + \int_0^{T_{k-1}-m} \alpha(t)dt = -\int_0^{T_k-m} b(t)dt - \int_0^{m-T_{k-1}} b(t)dt.
$$

Thus,

$$
\tilde{Q}(t_i = m) = \frac{N^{\frac{im}{T}}}{N^i(\ln N)^{i-1}} \exp\left(\int_0^{T+T_k-T_{k-1}} b(t)dt\right) \exp\left(-\int_0^{T_k-T_{k-1}} b(t)dt\right)
$$
$$
= \frac{N^{\frac{in}{T}}}{N^i(\ln N)^{i-1}} \exp\left(\int_0^T b(t)dt\right).
$$

**2**. When $m \geq T_k$: It's easy to show

$$
\int_0^{m-T_k} \alpha(t)dt + \int_0^{T_{k-1}-m} \alpha(t)dt = \int_0^{m-T_k} \left(c(t) - b(t)\right)dt - \int_0^{T_k - T_{k-1}} b(t)dt.
$$

Thus

$$
\tilde{Q}(t_i = m) = \frac{N^{\frac{im}{T}}}{N^i (\ln N)^{i-1}} \exp\left(\int_0^T b(t)dt\right) \exp\left(\int_0^{m-T_k} \left(c(t) - b(t)\right)dt\right).
$$

**3**. When $m \leq T_{k-1}$: It's easy to show

$$
\int_0^{m-T_k} \alpha(t)dt + \int_0^{T_{k-1}-m} \alpha(t)dt = \int_0^{T_{k-1}-m} \left(c(t) - b(t)\right)dt - \int_0^{T_k - T_{k-1}} b(t)dt,
$$

Thus

$$
\tilde{Q}(t_i = m) = \frac{N^{\frac{im}{T}}}{N^i (\ln N)^{i-1}} \exp\left(\int_0^T b(t)dt\right) \exp\left(\int_0^{T_{k-1}-m} \left(c(t) - b(t)\right)dt\right).
$$

The value of $Q(t_i = m)$ requires normalization. The normalization constant is given by:

$$
\begin{aligned}
C &= \int_0^N \tilde{Q}(t_i = m)dm \\
&= \mathcal{O}_p\left(\frac{N^{\frac{iT_k}{T}}}{N^i (\ln N)^{i-1}} \exp\left(\int_0^T b(t)dt\right)\right).
\end{aligned}
$$

Thus, the value of $Q(t_i = m) = \tilde{Q}(t_i = m)/C$ is given by:

$$
\begin{aligned}
Q(t_i = m) &= \begin{cases} N^{\frac{im - iT_k}{T}} & \text{if } m \in [T_{k-1}, T_k), \\ N^{\frac{im - iT_k}{T}} \exp\left(\int_0^{m-T_k} \left(c(t) - b(t)\right)dt\right) & \text{if } m \geq T_k, \\ N^{\frac{im - iT_k}{T}} \exp\left(\int_0^{T_{k-1}-m} \left(c(t) - b(t)\right)dt\right) & \text{if } m \leq T_{k-1}. \end{cases} \\
&= \begin{cases} 1 & \text{if } m = T_k, \\ \mathcal{O}_p(N^{\frac{m-T_k}{T}}) & \text{if } m \in [T_{k-1}, T_k), \\ \mathcal{O}_p(\exp(N^{\frac{\min\{|m-T_k|, |m-T_{k-1}|\}}{T}})) & \text{if } m \notin [T_{k-1}, T_k]. \end{cases}
\end{aligned}
$$

For junction points $T_i$ with $i \in \{M_k\}_{k=1}^K$, the unnormalized probability is given by:

$$
\tilde{Q}(t_i = m) = \frac{N^{\frac{in}{T}}}{N^i (\ln N)^{i-1}} \exp\left(\int_0^T b(t)dt\right) \exp\left(\int_0^{|T_{k-1}-m|} \left(c(t) - b(t)\right)dt\right).
$$

Then, the normalization constant is

$$
\begin{aligned}
C &= \int_0^N \tilde{Q}(t_i = m)dm \\
&= \mathcal{O}_p\left(\frac{N^{\frac{iT_k}{T}}}{N^i (\ln N)^{i-1}} \exp\left(\int_0^T b(t)dt\right)\right).
\end{aligned}
$$

Since $Q(t_i = m) = \tilde{Q}(t_i = m)/C$, we have

$$
\begin{aligned}
Q(t_i = m) &= N^{im - iT_k} \exp\left(\int_0^{|m-T_k|} \left(c(t) - b(t)\right)dt\right) \\
&= \mathcal{O}_p\left(\exp\left(-N^{\frac{|m-T_k|}{T}}\right)\right).
\end{aligned}
$$

Hence the proof is finished.

$\square$

# D    SIMULATION SETTINGS

## D.1    INITIALIZATION AND HYPERPARAMETERS SETTING

The hierarchical model given in **Equation** 1 has hyperparameters $\alpha = \{\beta, \nu^0, V^0\}$. To implement **Algorithm** 1, the hyperparameters in the conjugate prior defined in **Equation** 1 set as follows: the parameter $\beta$ in all Gaussian prior distributions $\mathcal{N}\left(0, \beta^{-1}I\right)$ are set to the data dimension $D$. Similarly the prior Wishart distribution $\mathcal{W}(\nu^0, V^0)$ is assigned with $\nu^0 = D, V^0 = D \cdot \mathbf{I}$ where $\mathbf{I}$ is identity matrix of dimension $D$. The Gaussian-Wishart prior has been studied for low-rank matrix completion, which imposes an appropriate penalty, and encourages sparse solutions with promising convergence.

Throughout all the experiments, the initialization of **Algorithm** 1 follows the description of **A3** in **Section** 2.3 where we evenly divide the entire sequence into $\tilde{K}$ segments. Then $\{Q(\theta)\}_{k=1}^{\tilde{K}}$ are initialized using the statistical moments (mean and variance for the Gaussian distribution) from these segments.

## D.2    NUMERICAL DEMONSTRATION

In this section, We evaluate the performance by varying values of sequence length $N$. In particular, we consider sampling the 1-dimensional mean-variance shift sequence with five equally spaced segments. The length of each segment is $N/5$ and $N$ varies in the set $\{50, 100, 200, 400, 600, 800, 1000, 1200, 1400, 1600\}$. In each segment, samples are drawn from a normal distribution with the following parameters.

$$\boldsymbol{u} = [0, 3, 2, 4, 4] \qquad \Lambda = [1, 0.25, 1, 1, 4]$$

Elements in $\boldsymbol{u}$ and $\Lambda$ represent the mean and precision of a particular segment. For example, samples in the first segment follow a $\mathcal{N}(0, 1)$ distribution. We initialize our algorithm with $\tilde{K} = 10$ and set the iteration number to 30. The simulations are repeated 100 times and the average number of change points and average mean absolute error are reported in Figure 2.

## D.3    LOCATION AND PARAMETER ESTIMATION

In this subsection, we consider one mixed distribution sequence and two normal sequences. Model 1 is a variance shift sequence model. The five ordered segments are sample from $Binomial(10, 0.3)$, $\mathcal{N}(3, 4)$, $Poisson(3)$, and $Binomial(15, 0.2)$, each with 100 samples. For Model 2 and Model 3, the multivariate normal sequences, parameters are specified in the following Table 3. Let

$$\boldsymbol{u}_1^{(2)} = [0, 0, 0, 0, 0], \quad \boldsymbol{u}_2^{(2)} = \boldsymbol{u}_3^{(2)} = [0, 2, 0, 1, 2], \quad \boldsymbol{u}_4^{(2)} = \boldsymbol{u}_5^{(2)} = [4, 0, 2, 0, 4],$$

$$\boldsymbol{u}_1^{(3)} = [\mathbf{0}_{10}], \quad \boldsymbol{u}_2^{(3)} = \boldsymbol{u}_3^{(3)} = [0, 2, 0, 1, 0, 1, 0, 0, 0, 1], \quad \boldsymbol{u}_4^{(3)} = \boldsymbol{u}_5^{(3)} = [1, 0, 2, 0, 4, 0, 0, 4, 0, 1].$$

$\mathbf{I}$ is the identity matrix of size $D$ and $\mathbf{I}_{0.8}$ is an identity matrix with the non-diagonal elements equal to 0.8. $\mathbf{0}_{10}$ is a zero vector in 10-d. Specifically, all three Models are subject to four change points occurring at $\tau = \{100, 200, 300, 400\}$, each representing a change in distribution. Clearly, Model 2 and 3 incorporate the mean shift or correlation shift around change points. For all three models, $N$, the length of the sequence is 500, and the upper bound of the number of change points $\tilde{K}$ is 10. The iteration number is set to 30. For each model, the repetition of simulation is 100 and the average Rand index of each model is reported in Table 1.

Table 3: Normal Mean correlation-Shift of Model 2 and 3

|  | $\boldsymbol{u}$ | $\Lambda$ | $\tau$ | $D$ | $N$ |
|---|---|---|---|---|---|
| Model 2 | $[\boldsymbol{u}_1^{(2)}, \boldsymbol{u}_2^{(2)}, \boldsymbol{u}_3^{(2)}, \boldsymbol{u}_4^{(2)}, \boldsymbol{u}_5^{(2)}]$ | $\mathbf{I}_{0.8}^{-1}, \mathbf{I}_{0.8}^{-1}, \mathbf{I}, \mathbf{I}, \mathbf{I}_{0.8}^{-1}$ | $100, 200, 300, 400$ | $5$ | $500$ |
| Model 3 | $[\boldsymbol{u}_1^{(3)}, \boldsymbol{u}_2^{(3)}, \boldsymbol{u}_3^{(3)}, \boldsymbol{u}_4^{(3)}, \boldsymbol{u}_5^{(3)}]$ | $\mathbf{I}_{0.8}^{-1}, \mathbf{I}_{0.8}^{-1}, \mathbf{I}, \mathbf{I}, \mathbf{I}_{0.8}^{-1}$ | $100, 200, 300, 400$ | $10$ | $500$ |

In this part, we evaluate the accuracy of the posterior parameter estimation. Here we only consider normal sequence cases for estimation. The parameters are summarized in Table 4

Table 4: Normal Mean correlation-Shift in Case 1, 2 and 3

|  | $\boldsymbol{u}$ | $\Lambda$ | $\tau$ | $D$ | $N$ |
|---|---|---|---|---|---|
| Case 1 | $[0, 3, 2, 4, 4]$ | $1, 0.25, 1, 1, 4$ | $100, 200, 300, 400$ | $1$ | $500$ |
| Case 2 | $[\boldsymbol{u}_1^{(2)}, \boldsymbol{u}_2^{(2)}, \boldsymbol{u}_3^{(2)}, \boldsymbol{u}_4^{(2)}, \boldsymbol{u}_5^{(2)}]$ | $\mathbf{I}_{0.8}^{-1}, \mathbf{I}_{0.8}^{-1}, \mathbf{I}, \mathbf{I}, \mathbf{I}_{0.8}^{-1}$ | $100, 200, 300, 400$ | $5$ | $500$ |
| Case 3 | $[\boldsymbol{u}_1^{(3)}, \boldsymbol{u}_2^{(3)}, \boldsymbol{u}_3^{(3)}, \boldsymbol{u}_4^{(3)}, \boldsymbol{u}_5^{(3)}]$ | $\mathbf{I}_{0.8}^{-1}, \mathbf{I}_{0.8}^{-1}, \mathbf{I}, \mathbf{I}, \mathbf{I}_{0.8}^{-1}$ | $100, 200, 300, 400$ | $10$ | $500$ |

The included symbols are the same as above. The estimation error (Mean square error) is measured by taking $l^2$ norm of the difference between the estimated mean and ground truth, while the MSE.SD is the ordinary standard deviation of estimation. Notice that in the estimation of the covariance matrix, the estimation error is further divided by data dimension $D$ to maintain numerical consistency. These simulations are also repeated 100 times and the average MSE is shown in Table 2:

$$\text{Mean square error } = \frac{1}{N} \sum_{i=1}^{N} \left\| \hat{\boldsymbol{\theta}}_i - \boldsymbol{\theta_0} \right\|_2^2, \tag{5}$$

and

$$\text{MSE.SD } = \sqrt{\frac{1}{N-1} \sum_{i=1}^{N} \left( \left\| \hat{\boldsymbol{\theta}}_i - \boldsymbol{\theta_0} \right\|_2^2 - \frac{1}{N} \sum_{j=1}^{N} \left\| \hat{\boldsymbol{\theta}}_j - \boldsymbol{\theta_0} \right\|_2^2 \right)^2}. \tag{6}$$

### D.4 NON-GAUSSIAN EXAMPLES SETTINGS

In these non-Gaussian examples, we consider testing the performance on Poisson, chi-squared, or the exponential random sequences. For the Poisson sequence, the rate parameters $\boldsymbol{\lambda}$ of five segments are $\boldsymbol{\lambda} = [1, 5, 2, 10, 3]$. $\mathbf{df} = [1, 5, 2, 4, 1]$ are set to be the parameters of chi-squared sequence. The scale parameters of the exponential distribution are $\boldsymbol{\beta} = [1, 5, 0.5, 4, 1]$. $N$, the length of the sequence is 500 and each segment contains 100 samples. $\tilde{K}$, the upper bound of the number of change points is 8.

