## A PRELIMINARY

### A.1 CHANGE POINT DETECTION METHODS

**WBSLSW**: The WBSLSW method (Korkas & PryzlewiczV, 2017) incorporates the non-parametric locally stationary wavelet process with wild binary segmentation, and can detect the second-order structure of the sequence with an unknown number of change points. We implement the WBSLSW using the R package `wbsts`.

**KCP**: The KCP (Harchaoui & Cappé, 2007) method is a dynamic programming method with a known number of change points. This algorithm detects the change points by minimizing the kernel least-squares criterion. In our cases, we combine KCP with a linear penalty pruning the number of change points. This is done by using the python package `ruptures` (Truong et al., 2020) with a Gaussian kernel and default parameters.

**DPHMM**: The DPHMM method developed by Ko et al. (2015) is a combination of the Dirichlet process and the Hidden Markov model to detect change points using MCMC. This method allows the number of change points to be unknown. We implement this algorithm using the R package `dirichletprocess` with default parameters and set $\tilde{K} = 10$

**ECP3O**: Zhang et al. (2017) propose a new change point search framework called change point procedure via pruned objectives. The ECP3O method uses the new search frame with the E-statistics which measures the goodness-of-fit. This method is implemented using `e.cp3o-delta` function with default parameters in R package `ecp`. (Nicholas A. James and Wenyu Zhang and David S. Matteson, 2019). The maximum number of change points $\tilde{K}$ is 10.

$\mathcal{D}_m$**-BOCD**: $\mathcal{D}_m$-BOCD is an online method developed by Altamirano et al. (2023). This method is generalized from BOCD (Adams & MacKay, 2007) with diffusion score matching, which is robust to sequences with outliers. We implement this method using the code provided on the author's GitHub page.

### A.2 VARIATIONAL INFERENCE

Variational inference (VI) Blei et al. (2017) works as a fast approximation method for Bayesian inference. Given the observation $\mathbf{x}$ and latent variable $\mathbf{z}$, VI uses a tractable variational distribution $q$ drawn from the function class $\mathcal{F}$ to approach the complicated posterior $p(\mathbf{z} \mid \mathbf{x})$ by minimizing their KL divergence. However, the KL can not be computed analytically. Classical VI optimizes an alternative objective called Evidence Lower Bound (ELBO) that is equivalent to log marginal likelihood minus the KL:

$$
\begin{aligned}
\ln p(\mathbf{x}) &= \text{ELBO}(q) + \text{KL}(q \| p(\mathbf{z} \mid \mathbf{x})) \\
&= \int q(\mathbf{z}) \ln \left\{ \frac{p(\mathbf{x}, \mathbf{z})}{q(\mathbf{z})} \right\} d\mathbf{z} - \int q(\mathbf{z}) \ln \left\{ \frac{p(\mathbf{z} \mid \mathbf{x})}{q(\mathbf{z})} \right\} d\mathbf{z}.
\end{aligned}
$$

Under the common mean-field assumption $q(\mathbf{z}) = \prod_{j=1}^{m} q_j(z_j)$, the maximizer of ELBO $q_j^*$ has the analytical solution $q_j^*(z_j) \propto \exp\left\{ \mathbb{E}_{\mathbf{z}_{-j}} \left[ \log p(\mathbf{z}, \mathbf{x}) \right] \right\}$, where $\mathbf{z}_{-j}$ denotes all variables $z_i$ other than $z_j$, that can be solved by the coordinate ascent algorithm.

Recently, VI has been commonly applied in training deep generative models, including VAE (Kingma & Welling, 2013) and deep diffusion model (Ho et al., 2020) to approximate complex posterior distributions. VI plays a crucial role in approximating the posterior distribution over the latent variables, enabling efficient learning and generation of high-quality samples from complex data distributions.

## B NORMAL MEAN-VARIANCE SHIFT MODEL

In this section, we derive updating formulas for the Normal Mean-Variance Shift model. Denoting the set of all latent variables as $\boldsymbol{\xi} = \{\{\mathbf{t}_k\}_{k=1}^{K}, \{\theta_{K+1}\}_{k=1}^{K+1}\}$ and the hyperparameters set $\alpha_k = $

$\{\beta, \nu_0, V_0\}$, we assume a constrained mean-field $Q$ family in variational inference:

$$Q(\boldsymbol{\xi}) = Q(\mathbf{t}_1) \prod_{k=2}^{K} Q(\mathbf{t}_k | \mathbf{t}_{k-1}) \prod_{k}^{K+1} Q(u_k) Q(\Lambda_k).$$

Then the variational lower bound is given by

$$
\begin{aligned}
\mathcal{L}(Q) = & \sum_{k-1}^{K+1} \left[ \sum_{t_k, t_{k-1}} Q(\mathbf{t}_k, \mathbf{t}_{k-1}) \int Q(u_k) Q(\Lambda_k) \ln p(\mathbf{Y}_k \mid \mathbf{t}_k, \mathbf{t}_{k-1}, u_k, \Lambda_k) \, du_k d\Lambda_k \right] \\
& + \sum_{k=1}^{K+1} \left[ \int Q(u_k) Q(\Lambda_k) \ln \frac{\mathcal{N}(u_k; 0, \beta^{-1} I) \mathcal{W}(\Lambda_k; \nu^0, V^0)}{Q(u_k) Q_T(\Lambda_k)} du_k d\Lambda_k \right] \\
& + \sum_{k=1}^{K} \left[ \sum_{t_k, t_{k-1}} Q_t(\mathbf{t}_k, \mathbf{t}_{k-1}) \ln \frac{p(\mathbf{t}_k \mid \mathbf{t}_{k-1})}{Q_t(\mathbf{t}_k \mid \mathbf{t}_{k-1})} \right].
\end{aligned}
$$

Minimizing KL divergence leads to an analytical solution. We can directly apply it to give the optimal solutions for the family of factors $Q$ of variational posteriors

$$Q(\mathbf{t}_1) = \prod_{i=1}^{N} \tilde{\pi}_{1,i}^{\mathbf{t}_1(i)}, \quad Q(\mathbf{t}_k | \mathbf{t}_{k-1}) = \prod_{i=1}^{N} \prod_{j=1}^{N} \hat{\pi}_{k,i,j}^{\mathbf{t}_k(i) \times \mathbf{t}_{k-1}(j)},$$

$$Q(u_k) = \mathcal{N}(u_k \mid m_k, L_k^{-1}), \quad Q(\Lambda_k) = \mathcal{W}(\Lambda_k \mid \nu_k, V_k).$$

Given prior distributions defined above, solutions for variational parameters are given by

$$
\begin{aligned}
m_k = & \left[ \langle \Lambda_k \rangle \sum_{n=1}^{N} \sum_{m \geq n}^{N} Q(\mathbf{t}_k(m), \mathbf{t}_{k-1}(n)) \sum_{j=n}^{m} \mathbf{1} + \mathbf{I}/\beta \right]^{-1} \\
& \times \left[ \langle \Lambda_k \rangle \sum_{n=1}^{N} \sum_{m \geq n}^{N} Q(\mathbf{t}_k(m), \mathbf{t}_{k-1}(n)) \sum_{j=n}^{m} y_j \right], \\
L_k = & \langle \Lambda_k \rangle \sum_{n=1}^{N} \sum_{m \geq n}^{N} Q(\mathbf{t}_i(m), \mathbf{t}_{i-1}(n)) \sum_{j=n}^{m} \mathbf{1} + \alpha^{-1} \mathbf{I}, \\
\nu_i = & \nu_0 + \sum_{n=1}^{N} \sum_{m \geq n}^{N} Q(\mathbf{t}_i(m), \mathbf{t}_{i-1}(n)) \sum_{j=n}^{m} \mathbf{1},
\end{aligned}
$$

and

$$V_k^{-1} = V_0^{-1} + \sum_{n=1}^{N} \sum_{m \geq n}^{N} Q(\mathbf{t}_k(m), \mathbf{t}_{k-1}(n)) \left[ \sum_{j=n}^{m} y_j y_j^\top - 2 \sum_{j=n}^{m} y_j \langle u_k \rangle^\top + \sum_{j=n}^{m} \langle u_k u_k^\top \rangle \right].$$

As we mentioned in **Section** 2.2, solutions for $Q_t(\mathbf{t}_k \mid \mathbf{t}_{k-1})$ can be obtained through sum-product algorithm. The updating formulas are given by

$$Q(\mathbf{t}_k(n) = 1) = \mu_{\rightarrow \mathbf{t}_k}(n) \cdot \mu_{\mathbf{t}_k \leftarrow}(n) = \tilde{\pi}_{k,n},$$

and

$$
\begin{aligned}
& Q(\mathbf{t}_{k-1}(m) = 1, \mathbf{t}_k(n) = 1) \\
& = \mu_{\rightarrow \mathbf{t}_{k-1}}(m) \cdot \pi_{i,m,n} \cdot \exp\left( \mathrm{E}_{Q(u_k) Q(\Lambda_k)} \ln p(\mathbf{Y}_k, \mathbf{t}_k, u_k, \Lambda_k \mid \mathbf{t}_{k-1}) \right) \cdot \mu_{t_k \leftarrow}(n),
\end{aligned}
$$

where messages are obtained recursively. For $k = 1, \dots, K$,

$$
\begin{aligned}
\mu_{\rightarrow \mathbf{t}_k}(n) = & \sum_{m=1}^{n} \left\{ \mu_{\rightarrow \mathbf{t}_{k-1}}(m) \cdot \pi_{k,m,n} \cdot \exp\left\{ \sum_{j=m}^{n} \frac{1}{2} \langle \ln |\Lambda_k| \rangle \right. \right. \\
& \left. \left. - \frac{1}{2} \sum_{j=m}^{n} \mathrm{Tr}\left[ (y_j \cdot y_j^\top - y_j \cdot \langle u_k^\top \rangle - \langle u_k \rangle \cdot y_j^\top + \langle u_k, u_k^\top \rangle) \cdot \langle \Lambda_k \rangle \right] \right\} \right\},
\end{aligned}
$$

and

$$
\begin{aligned}
\mu_{\mathbf{t}_{k-1}\leftarrow}(m) \;=\; & \sum_{n=m}^{N}\Bigg\{ \mu_{\mathbf{t}_{k-1}\leftarrow}(n)\cdot\pi_{k,m,n}\cdot\exp\Bigg\{\sum_{j=m}^{n}\frac{1}{2}\left\langle\ln|\Lambda_k|\right\rangle \\
& -\frac{1}{2}\sum_{j=m}^{n}\mathrm{Tr}\left[\left(y_j\cdot y_j^{\top}-y_j\cdot\left\langle u_k^{\top}\right\rangle+\left\langle u_k,u_k^{\top}\right\rangle\right)\cdot\left\langle\Lambda_k\right\rangle\right]\Bigg\}\Bigg\}.
\end{aligned}
$$

To start recursion, the initial message state $\mu_{\to\mathbf{t}_1}$ and $\mu_{\mathbf{t}_K\leftarrow}$ are given by

$$
\begin{aligned}
\mu_{\to\mathbf{t}_1}(m) \;=\; & \pi_{1,m}\exp\Bigg\{\sum_{j=m}^{n}\frac{1}{2}\left\langle\ln|\Lambda_1|\right\rangle \\
& -\frac{1}{2}\sum_{j=m}^{n}\mathrm{Tr}\left[\left(y_j\cdot y_j^{\top}-y_j\cdot\left\langle u_1^{\top}\right\rangle-\left\langle u_1\right\rangle\cdot y_j^{\top}+\left\langle u_1,u_1^{\top}\right\rangle\right)\cdot\left\langle\Lambda_1\right\rangle\right]\Bigg\},
\end{aligned}
$$

and

$$
\begin{aligned}
\mu_{\mathbf{t}_K\leftarrow}(m) \;=\; & \exp\Bigg\{\sum_{j=m}^{n}\frac{1}{2}\left\langle\ln|\Lambda_{K+1}|\right\rangle \\
& -\frac{1}{2}\sum_{j=m}^{n}\mathrm{Tr}\left[\left(y_j\cdot y_j^{\top}-y_j\cdot\left\langle u_{K+1}^{\top}\right\rangle+\left\langle u_{K+1},u_{K+1}^{\top}\right\rangle\right)\cdot\left\langle\Lambda_{K+1}\right\rangle\right]\Bigg\},
\end{aligned}
$$

where we have assumed:

$$
\left\langle u_k\right\rangle=m_k,\quad \left\langle u_k u_k^{\top}\right\rangle=m_k m_k^{\top}+L_k^{-1},\quad \left\langle\Lambda_k\right\rangle=\nu_k V_k,
$$

$$

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

_1^0 \right\rangle^\top - \left\langle u_1^0 \right\rangle \cdot y(t)^\top + \left\langle u_1^0, u_1^{0\top} \right\rangle \right) \cdot \left\langle \Lambda_1^0 \right\rangle \right] \right\} \right\}$$

$$= \begin{cases} \mathcal{O}_p(\pi_{1,m} \cdot \exp \left\{ \int_0^m b(t)dt \right\}) & \text{if } m \leq T_1, \\[2mm] \mathcal{O}_p(\pi_{1,m} \cdot \exp \left\{ \int_0^{T_1} b(t)dt + \int_{T_1}^m c(t)dt \right\} & \text{if } m \geq T_1. \end{cases}$$

$$= \mathcal{O}_p \left( \frac{1}{N} \exp \left( \int_0^{T_1} b(t)dt \right) \cdot \exp \left( \int_{T_1}^m \alpha(t)dt \right) \right),$$

where we use the fact that $\pi_{i,m} = 1/N$ and the initial segment is a subset of the first regime $[0, t_1] \subset [0, T_1]$ and the initialized parameters are consistent with the true value $S_1$ and $u_1$, such that:

$$\left\langle u_1^0 \right\rangle = \hat{u}_1 \xrightarrow{\mathrm{P}} u_1, \quad \left\langle u_1^0, u_1^{0\top} \right\rangle = \hat{u}_1 \cdot \hat{u}_1^\top, \quad \left\langle \Lambda_1^0 \right\rangle = \hat{S}_1 \xrightarrow{\mathrm{P}} S_1, \quad \left\langle \ln \left| \Lambda_1^0 \right| \right\rangle = \ln \left| \hat{S}_1 \right| \xrightarrow{\mathrm{P}} \ln \left| S_1 \right|.$$