# OpenReview forum: "Change Point Detection via Variational Time-Varying Hidden Markov Model"
_ICLR.cc/2024/Conference — Submitted to ICLR 2024_

### Official Review · Reviewer_bXnv · 2023-10-30

**Soundness:** 2 fair
**Presentation:** 2 fair
**Contribution:** 3 good
**Rating:** 6
**Confidence:** 4

**Summary:**

This paper tackles the problem of change point detection in the offline setting. While a large span of previous methods rely on Hidden Markov Models, the authors introduce TV-HMM, a variant of the Hidden Markov Model incorporating the time-varying location transition matrix. An EM-based algorithm is proposed for inference with theoretical guarantees. The TV-HMM model is shown to lead to more robust results compared to standard HMMs and is better suited when the number of change points is not known.

An extension of the TV-HMM model to a semi-parametric setting is proposed, getting rid of the usual restrictive distribution assumptions.

**Strengths:**

- The authors propose a different viewpoint on the change point detection problem. As far as I know, their approach is new and allows to obtain more robust results compared to standard methods when the number of change points is not known.

- The authors compare their approach with other benchmark methods and show the good performance and the robustness of their method.

- The authors propose an interesting extension of their model to bypass the restrictive parametric assumption on the distribution of the observations.

**Weaknesses:**

- The current version contains a few typographical errors and some notational issues, which make it somewhat challenging to read.

- It seems that the results and details of the simulations corresponding to Section 4 (the semi-paramteric model) are not given in the paper. (In particular, I would have been curious to know how the author select the mapping $\phi$ in their simulations.)

- A more detailed comparison with other methods (particularly in terms of computer complexity) would have been useful.

**Questions:**

I thank the authors for this nice submission. Some of my questions are already listed in the previous sections. My others questions/comments (including some typos) are presented below.

- I appreciate the discussion on computational complexity at the end of Section 2.2. Nevertheless, I would have been interested to see in the paper a more detailed discussion about the computational complexity of the algorithm (in terms of time and space) compared to others approaches.

- A closing parenthesis is missing at page 4 in the E-step equation. Same in the first line of Eq.(2).

- In assumption A1, I think in the Gaussian distribution $S_k$ should be $\Lambda_k$ (to be consistent with the notations used in Eq.(1)).

- There is a typo in the use of the notation $M_{K+1}$. For example, in the first bullet point of Assumption A3, it should be $t_i\in \{1,\dots, M_{k+1}-1\}$. (Let me point out that $M_k$ for $k\neq K+1$ have not been defined and I think that they should not be used anyway). The same holds for the second bullet point of assumption A3, Theorem 1, Figure 1 and Section C of the Appendix.

- In Section 4, the use of the notation $\theta_k$ is confusing. Indeed, before section 4, we work in the full parametric case and $\theta_k=(u_k,\Lambda_k)$ (the parameter of the Gaussian distribution from which points are independently sampled from after the $k$-th change point). However, $\theta_k$ in Section 4 needs to be a vector in $\mathbb R^D$ (i.e. in the same space as the observations). Therefore, it might be good to use another notation or to stress at the beginning of Section 4 the properties of $\theta_k$.

- In Eq.(4), the parameters $\pi_{k,m,n}$ are still considered but the update from line 8 (Algo 1) is not presented anymore in Algo 2. I would be grateful if the authors could clarify this point.

- Below Algorithm 2, there should not be a final point: "Unlike Equation 2 in the parametric model, where $Q(\theta)$ must be derived using variational inference. Here, $Q(\theta)$ can be generally modeled using non-parametric density estimation...". Furthermore, using "must" suggests that no alternative can be considered. However, I would say that standard MCMC techniques could be used instead of variational inference. If the authors agree, I would suggest to reformulate the sentence. In the second sentence, it is also mention that $Q$ can be modeled using non parametric techniques, but the following sentence (and the presented algorithm) only deal with a parametric distribution (with parameter $\Phi$). It might be good to mention how the algorithm is changed when a non-parametric density estimation method is used.

- To be consistent with the notations introduced in sections 2 and 3, should $K$ in Algorithm 2 be replaced by $\tilde K$ (since $K$ is the true and unknown number of change points) ?

- In Section 4, "from MMD-based message passing of Equaton 4..." : it should be "Equation".

---

> ### Author Response · Authors · 2023-11-22
>
> Dear Reviewer,
>
> Thank you very much for your comments. \
> \
> Q1: “There is a typo in the use of the notation M_K+1. For example, in the first bullet point of Assumption A3, it should be $t_i \in1 ,\ldots, M_{k+1} -1$. (Let me point out that
>  for $k \neq K+1$ have not been defined and I think that they should not be used anyway). The same holds for the second bullet point of assumption A3, Theorem 1, Figure 1 and Section C of the Appendix. ”
>
> Thank you very much for raising this question. In the initialization, $M_{K+1}$ is the number of regimes in the sequence, so there are $M_{K+1}-1$ initialized change points. Suppose there are $K$ true change points, $t_{M_k}$ for $k \neq K+1$ is defined as the initialized point closest to $T_k$, where $t_{M_k}$ is specifically located on the right side of $T_k$. As stated in A3, the interval $[t_{M_k -1}, t_{ M_k }]$ covers the change point $T_k$, and points in this interval are not identically distributed. \
> Notation: $T_k$ is the $k$-th true change point, $k \in 1,\ldots,K$\
> $t_{M_k}$ is the right closest initialized point of $T_k$, $k \in 1,\ldots,K$\
> $t_{M_{k-1}}$ is the right closest initialized point of $T_{k-1}$\
> $t_{M_k-1}$ is the adjacent initialized point of $t_{M_k}$ on the left.\
> \
> Q2: “In Eq.(4), the parameters \pi are still considered but the update from line 8 (Algo 1) is not presented anymore in Algo 2. I would be grateful if the authors could clarify this point. ”\
> \
> Indeed, the parameters $\pi$ still need to be considered in Algorithm 2, and we have added one more step for $\pi$ in the Algorithm.\
> \
> Q3: “Furthermore, using "must" suggests that no alternative can be considered. However, I would say that standard MCMC techniques could be used instead of variational inference. If the authors agree, I would suggest to reformulate the sentence. In the second sentence, it is also mention that \Q can be modelled using non parametric techniques, but the following sentence (and the presented algorithm) only deal with a parametric distribution (with parameter ). It might be good to mention how the algorithm is changed when a non-parametric density estimation method is used.”\
> \
> Thank you very much for your suggestion. Here we use "must" because $Q(\theta)$ with explicit form in Algorithm 1 is developed under the framework of Variational Inference. Although we can use MCMC to perform inference on parametric TV-HMM, the computational cost is higher. The main difference between the non-parametric and parametric TV-HMM is that we substitute the parametric likelihood measurement with the Maximum Mean Discrepancy(MMD). We first sample $\tilde K+1$  sequences from non-parametrically estimated $Q(\zeta)$. Each $Q(\zeta_k)$ represents a distinguished regime distribution of the sequence. Calculating the MMD between the sampled sequences and the original sequence, we can find the best-matching segment to determine the change point location. Although the parametric model assumption can help derive analytic solutions and enhance the convergence rate, the model may become compromised if the assumption is violated. e.g., the DPHMM method is not well performed on a sequence with piecewise linear segments.\
> \
> Q4: “The current version contains a few typographical errors and some notational issues, which make it somewhat challenging to read.”\
> \
> Thank you very much for the correction. We’re sorry about the typographical errors and the notational issues in the paper and have already corrected them. We have changed all $S_k$ into $\Lambda_k$ and replaced $K$ with $ \tilde K$ in Algorithm 2, for consistent usage of symbols. The $\theta_k$ in Section 4 is substituted with $\zeta_k$ to avoid confusion.

---

> > ### Author Response · Authors · 2023-11-23
> >
> > Experiment:\
> > \
> > We also compared the semi-parametric TV-HMM with DPHMM on a sequence with piecewise linear segments to show the improvement. In this experiment, the data are generated from the model $\beta_0+\beta_1*t+\epsilon$, $\epsilon$ follows a normal distribution with mean 0 and variance 1. The length of the sequence is 600 with 4 inequal distanced change points. The change point locations are $\tau=[ 80 , 230, 330, 480]$ indicating lengths of each segments are $[ 80,150,100,150,120]$. $\tilde{K}$ is 10. The $\beta_0$ and $\beta_1$ are shown in the following table.
> >
> > Model: $y(t)=\beta_0+\beta_1*t+\epsilon$,  $\epsilon \sim N(0,1)$\
> > Total sample size: $N=600$\
> > Change point location: $\tau=[ 80 , 230, 330, 480]$\
> > Initialized number of change point: $\tilde{K}=10$
> >
> > |           | Segment 1 | Segment 2 |  Segment 3 |  Segment 4 | Segment 5 |
> > |----------:|:---------:|:---------:|:----------:|:----------:|:----------:|
> > | $\beta_0$ |     0     | $seg_1(80)$ |$ seg_2(150)$ | $seg_3(100)$ | $seg_4(150)$ |
> > | $\beta_1$ |    0.1    |    -0.3   |     0.3    |    -0.2    |    -0.2    |
> >
> >  $seg_i(j)$ stands for the $j$-th element in $i$-th segment.\
> > \
> > The summarized average Rand Index and its standard deviation are shown in the following table. It can be observed that Semi-parametric TV-HMM outperforms the other compared methods, indicating that our proposed method can handle the change point detection problem in sequences with linear trends. Moreover, the standard deviation of the Randx Index measurement is 0.02512, demonstrating the stability of our method.
> > |            |   semi-TVHMM   |      DPHMM     | ECP3O       |           WBSLSW |       KCP      |   $D_m$-BOCD   |
> > |-----------:|:--------------:|:--------------:|----------------|:--------------:|:--------------:|:--------------:|
> > | Rand Index |      0.9533    |      0.7882    |      0.8580    |      0.2209    |      0.8199    |      0.7637    |
> > |     SD     |     0.02512    |     0.06822    |     0.00422    |     0.05569    |     0.00197    |     0.09845    |

---

> > > ### Author Response · Authors · 2023-11-23
> > >
> > > Dear Reviewer,\
> > > \
> > > As the discussion period comes to an end, we are grateful to receive your insightful feedback on whether our response adequately addresses your concerns and questions.\
> > > \
> > > Thank you very much for your time and consideration.

---

### Official Review · Reviewer_98Hf · 2023-10-30

**Soundness:** 2 fair
**Presentation:** 2 fair
**Contribution:** 2 fair
**Rating:** 3
**Confidence:** 3

**Summary:**

The paper proposes a change point detection approach based on a time-varying Hidden Markov Model. The paper introduces a Hidden Markov Model with a time-varying location transition matrix and a corresponding inference method for this model based on variational expectation maximization (EM). The aim of the proposed method is the ability to deal with an undefined number of change points and, therefore, robustness against a mis specified number of change points. Furthermore, stochastic approximation allows to ease computation burden. Finally, the proposed approach operates within the common piece-wise i.i.d. setting and does not necessarily assume Gaussian likelihood.

**Strengths:**

The benefits of the approach are in its flexibility, in particular, the ability to learn the number of change points and flexibility in terms of likelihood specification.

**Weaknesses:**

Overall, the paper still needs improvement in clarity and more convincing experimental setup, which clearly shows in which situations the proposed approach would be preferred over existing methods and how it compares to other approaches in terms of computational speed.

**Questions:**

Detailed comments:
-	At times, writing is not always clear. For example, it is unclear what is the role of section 4 in the paper as there is no experimental evaluation of this extension. There are quite a few typos.

-	With the current details, I would struggle to implement the method and reproduce the results of the paper. I would suggest writing an additional section with implementation details, which ensures reproducibility.

-	The experimental setup is limited. In particular, the simulation study includes equally spaced change points, which is quite simplistic.

-	There is no computational comparison between the methods, and therefore, it is unclear how much one has to sacrifice in terms of speed to gain a little extra performance.

-	There are somewhat marginal differences in the performance between the proposed approach and other methods (both when the proposed approach underperforms and overperforms). From the current evaluation, it is unclear in what scenarios it would be beneficial to use the proposed method.

-	Table 1 lacks standard deviations over the 100 runs.

-	Some references do not include journal information or arxiv identifier.

- Figures are sometimes missing labels and captions of the figures/tables are not self-contained.

---

> ### Author Response · Authors · 2023-11-22
>
> Dear Reviewer,
>
> Thank you very much for your comments.\
> \
> Q1: “With the current details, I would struggle to implement the method and reproduce the results of the paper. I would suggest writing an additional section with implementation details, which ensures reproducibility.”\
> \
> Thank you very much for your advice. We have summarized all the experiment settings in Appendix D. The input parameters such as the initialized number of change points $\tilde K$, the sample sequence $Y$, and how it is generated are all included. We also add an additional experiment about the piecewise linear sequence. We are delighted to release the code after entering the final revision. Thank you again for your understanding and support.\
> \
> Q2: “There is no computational comparison between the methods, and therefore, it is unclear how much one has to sacrifice in terms of speed to gain a little extra performance.”\
> \
> The novelty is about extending the Hidden Markov Model. Since our proposed method is based on Variational Inference, the computational speed is indeed faster than previous DPHMM using MCMC sampling. The computational complexity for our proposed method is $\mathcal{O}(KN^2)$. If equipped with stochastic approximation, the complexity can be reduced to $\mathcal{O}(KS^2)$, and $S$ is the sub-sample size. The computational speed we can achieve for each step during iteration is approximately 1.73 seconds. \
> \
> Q3: “Table 1 lacks standard deviations over the 100 runs. Some references do not include journal information or arxiv identifier. Figures are sometimes missing labels and captions of the figures/tables are not self-contained.”\
> \
> Thank you very much for pointing out these issues. We have added the standard deviations over 100 runs and included the required information. We also check the reference section to ensure each reference includes journal information or arxiv identifier. Missing labels in the figures are fixed and captions are adjusted.
> |            |     Model 1     |     Model 2     |      Model 3     |
> |:----------:|:---------------:|:---------------:|:----------------:|
> |   WBSLSW   | 0.9068(0.08744) | 0.3596(0.22447) |  0.3849(0.23038) |
> |    ECP30   | 0.9156(0.04346) | 0.9580(0.05754) |  0.9737(0.01215) |
> | DPHMM      | 0.9637(0.02257) | 0.8727(0.03066) | 0.8869(0.02651)  |
> | KCP        | 0.9501(0.05622) | 0.8436(0.01631) | 0.8836(0.04664)  |
> | $D_m$-BOCD | 0.8123(0.11902)  | 0.8411(0.01894) | 0.8413(0.02208) |
> | TV-HMM     | 0.9523(0.05174) | 0.9756(0.01339) | 0.9615(0.02004)  |
>
> \
> \
> Experiment:\
> \
> \
> Thank you very much for your suggestion. We also test the semi-parametric TV-HMM on a sequence with piecewise linear segments to show the improvement. In this experiment, the data are generated from the model $\beta_0+\beta_1*t+\epsilon$, $\epsilon$ follows a normal distribution with mean 0 and variance 1. The length of the sequence is 600 with 4 inequal distanced change points. The change point locations are $\tau=[ 80 , 230, 330, 480]$ indicating lengths of each segments are $[ 80,150,100,150,120]$. $\tilde{K}$ is 10. The $\beta_0$ and $\beta_1$ are shown in the following table.
>
> Model: $y(t)=\beta_0+\beta_1*t+\epsilon$,  $\epsilon \sim N(0,1)$\
> Total sample size: $N=600$\
> Change point location: $\tau=[ 80 , 230, 330, 480]$\
> Initialized number of change point: $\tilde{K}=10$
>
> |           | Segment 1 | Segment 2 |  Segment 3 |  Segment 4 | Segment 5 |
> |----------:|:---------:|:---------:|:----------:|:----------:|:----------:|
> | $\beta_0$ |     0     | $seg_1(80)$ |$ seg_2(150)$ | $seg_3(100)$ | $seg_4(150)$ |
> | $\beta_1$ |    0.1    |    -0.3   |     0.3    |    -0.2    |    -0.2    |
>
>  $seg_i(j)$ stands for the $j$-th element in $i$-th segment.\
> \
> The summarized average Rand Index and its standard deviation are shown in the following table. It can be observed that Semi-parametric TV-HMM outperforms the other compared methods, indicating that our proposed method can handle the change point detection problem in sequences with linear trends. Moreover, the standard deviation of the Randx Index measurement is 0.02512, demonstrating the stability of our method.
> |            |   semi-TVHMM   |      DPHMM     |     ECP3O    |      WBSLSW      |       KCP      |   $D_m$-BOCD   |
> |-----------:|:--------------:|:--------------:|----------------|:--------------:|:--------------:|:--------------:|
> | Rand Index |      0.9533    |      0.7882    |      0.8580    |      0.2209    |      0.8199    |      0.7637    |
> |     SD     |     0.02512    |     0.06822    |     0.00422    |     0.05569    |     0.00197    |     0.09845    |

---

> > ### Author Response · Authors · 2023-11-23
> >
> > Dear Reviewer,\
> > \
> > As the discussion period comes to an end, we are grateful to receive your insightful feedback on whether our response adequately addresses your concerns and questions.\
> > \
> > Thank you very much for your time and consideration.

---

### Official Review · Reviewer_MqQg · 2023-10-31

**Soundness:** 2 fair
**Presentation:** 2 fair
**Contribution:** 3 good
**Rating:** 5
**Confidence:** 3

**Summary:**

The paper addresses the change point detection problem with a time-varying hidden Markov model. The authors develop a variational inference algorithm for parameter estimation as well as a semi-parametric extension using the Maximum Mean Discrepancy (MMD).

**Strengths:**

1. The model formulation is natural for the change point detection problem.
2. The authors generalize the log-likelihood based estimation with MMD to achieve improved robustness against model misspecification and outliers.
3. Theoretical guarantees are provided for the consistency of parameter estimation.

**Weaknesses:**

1. Placing the ARD prior on the elements of the transition matrix \Pi does not seem to be correct. How does this guarantee that the elements are nonnegative and add up to one? Typically, a Dirichlet prior is used which also induces sparsity. How does the proposed method perform compared to a Dirichlet prior?
2. Similarly the number of change points is determined by examining the posterior of the transition matrix. This also suffers from the issue above.
3. Is the left-to-right Markov chain assumption necessary? There could be identical regimes and the HMM can switch to a previously observed regime.
4. The authors adopted a SGD-type update for the posterior, e.g., line 8 of Algorithm 1. The update does not respect the sum to one constraint for the transition matrix.

**Questions:**

- How is the transition matrix posterior updated with the ARD prior? Specifically, why line 8. of Algorithm 1 ensures that the updated estimates are in a simplex?
- How does the proposed method perform compared to using Dirichlet priors on \Pi?
Minor:
\tau is used to denote change points (Sec 2) and also the step size in Algorithm 1.

---

> ### Author Response · Authors · 2023-11-22
>
> Dear Reviewer,
>
> Thank you very much for your comments.\
> \
> Q1: “How is the transition matrix posterior updated with the ARD prior? Specifically, why line 8. of Algorithm 1 ensures that the updated estimates are in a simplex?” \
> \
> The prior is initialized using an upper triangular matrix. In each row, entries are assigned with equal probability summing up to 1. Based on the concept of ARD prior, we investigate that the optimal $\Pi$ after differentiating the ELBO with respect to $\Pi$ is  the transition probability of the previous step $Q^S(t_k(n)=1|t_{k-1}(m)=1)$, which ensures the update estimates are in a simplex\
> \
> Q2: “How does the proposed method perform compared to using Dirichlet priors on \Pi? Minor: \tau is used to denote change points (Sec 2) and also the step size in Algorithm 1.”\
> \
> Thank you very much for pointing out the symbol issue. We have changed the symbol of the step size of $\tau$ into $\eta$. As mentioned before, each row of the initialized prior is uniformly distributed, with equal probabilities assigned to each element within the row. Our proposed algorithm can update the $\Pi_k$ iteratively based on the data. The specific entry $\pi_{k,m,n}$ of the ARD prior $\Pi_k$ measures the importance of the $k$-th change point transitioning from position $m$ to position $n$, which can be inferred from the data. The central concept of ARD prior is that the hyperparameters (in our cases, $\Pi_k$) are typically optimized by maximizing the marginal likelihood of the data.  Applying Dirichlet priors to $\Pi_k$ can yield similar results to iteratively updating $\Pi_k$. However, this approach might diminish the interpretability of  $\Pi_k$ , as $\pi_{k,m,n}$ becomes a random variable and can no longer be interpreted as a measure of importance or relevance. Moreover, when $\pi_{k,m,n}$ is treated as a random variable, it becomes less straightforward to visualize the sparsity of $\Pi_k$ directly in the plot.

---

> > ### Author Response · Authors · 2023-11-23
> >
> > Dear Reviewer,\
> > \
> > As the discussion period comes to an end, we are grateful to receive your insightful feedback on whether our response adequately addresses your concerns and questions.\
> > \
> > Thank you very much for your time and consideration.

---

### Official Review · Reviewer_hP3N · 2023-11-01

**Soundness:** 3 good
**Presentation:** 3 good
**Contribution:** 3 good
**Rating:** 5
**Confidence:** 3

**Summary:**

The paper addresses modeling time series data with sudden regime shifts, noting the limitations of the widely-used Hidden Markov Model (HMM). Introducing the TV-HMM, a variation with a time-varying location transition matrix, the authors offer a novel variational EM algorithm that pinpoints change point locations and quantities. This method remains robust against misidentification of change point numbers and has optimized computational efficiency. Statistical consistency under the Gaussian likelihood is assured, and a semi-parametric TV-HMM, free from distribution constraints, is also proposed.

**Strengths:**

1. The proposed variational EM algorithm is designed to be resilient against misidentification of change point numbers, and through the integration of stochastic approximation techniques, the paper addresses the computational intensity traditionally associated with HMMs.

2. The paper not only ensures the statistical consistency of change point location estimation under the Gaussian likelihood but also broadens its application by introducing a semi-parametric TV-HMM, which operates without stringent distribution assumptions, enhancing its adaptability to diverse data sets.

**Weaknesses:**

1. While the paper does propose a semi-parametric model free from stringent distribution assumptions, a significant portion of the study, including the assurance of statistical consistency, is still based on the Gaussian likelihood assumption, which may not always be applicable in real-world scenarios.

2. No improvement with respect to competitors.

**Questions:**

1. A clear discussion about the assumptions would be helpful.

2. Is assumption 2 strong with respect to literature?

3. The simulation study does not show an improvement. I think it would be helpful to see a more substantial improvement.

---

> ### Author Response · Authors · 2023-11-22
>
> Dear Reviewer,
>
> Thank you very much for your comments. \
> \
> Q1&2: "A clear discussion about the assumptions would be helpful. Is assumption 2 strong with respect to literature? "\
> \
> Thank you very much for your advice. The assumption A1 describes the order of each change point, and the last change point is fixed at $T$ where the first change point $T_1$ is after 0. We also assume the data $y$ in different segments are drawn from a normal distribution with different parameters depending on the regimes. For assumption A2, we assume the number of observations within a specific interval is of polynomial order in the total number of samples, N. Lastly, assumption A3 is about the characteristics of initialized change points. Equally distanced initial change points can be separated into junction and non-junction points. The junction point $t_i$ means there is a true change point within the interval $[t_{i-1},t_i]$. Junction and non-junction points are also demonstrated in Figure 1.
>
> While the number of observations within an interval is assumed to be polynomially related to the total sample N, this assumption is considered acceptable for developing the algorithms. Let $\alpha = (n-m)/T$, clearly $\alpha$ is less than or equal to 1, which means the rate is slower than N as N approaches infinity. For example, Chakar et al. (2017) prove their change point consistency as N approaches infinity. Thus, our assumption A2 is mild with respect to literature.
>
>
> Reference
>
> S. Chakar. E. Lebarbier. C. Lévy-Leduc. S. Robin. "A robust approach for estimating change-points in the mean of an AR(1) process." Bernoulli 23 (2) 1408 - 1447, May 2017\
> \
> \
> \
> Experiment:\
> \
> \
> Thank you very much for your suggestion. We also test the semi-parametric TV-HMM on a sequence with piecewise linear segments to show the improvement. In this experiment, the data are generated from the model $\beta_0+\beta_1*t+\epsilon$, $\epsilon$ follows a normal distribution with mean 0 and variance 1. The length of the sequence is 600 with 4 inequal distanced change points. The change point locations are $\tau=[ 80 , 230, 330, 480]$ indicating lengths of each segments are $[ 80,150,100,150,120]$. $\tilde{K}$ is 10. The $\beta_0$ and $\beta_1$ are shown in the following table.
>
> Model: $y(t)=\beta_0+\beta_1*t+\epsilon$,  $\epsilon \sim N(0,1)$\
> Total sample size: $N=600$\
> Change point location: $\tau=[ 80 , 230, 330, 480]$\
> Initialized number of change point: $\tilde{K}=10$
>
> |           | Segment 1 | Segment 2 |  Segment 3 |  Segment 4 | Segment 5 |
> |----------:|:---------:|:---------:|:----------:|:----------:|:----------:|
> | $\beta_0$ |     0     | $seg_1(80)$ |$ seg_2(150)$ | $seg_3(100)$ | $seg_4(150)$ |
> | $\beta_1$ |    0.1    |    -0.3   |     0.3    |    -0.2    |    -0.2    |
>
>  $seg_i(j)$ stands for the $j$-th element in $i$-th segment.\
> \
> The summarized average Rand Index and its standard deviation are shown in the following table. It can be observed that Semi-parametric TV-HMM outperforms the other compared methods, indicating that our proposed method can handle the change point detection problem in sequences with linear trends. Moreover, the standard deviation of the Randx Index measurement is 0.02512, demonstrating the stability of our method.
> |            |   semi-TVHMM   |      DPHMM     |     ECP3O    |      WBSLSW      |       KCP      |   $D_m$-BOCD   |
> |-----------:|:--------------:|:--------------:|----------------|:--------------:|:--------------:|:--------------:|
> | Rand Index |      0.9533    |      0.7882    |      0.8580    |      0.2209    |      0.8199    |      0.7637    |
> |     SD     |     0.02512    |     0.06822    |     0.00422    |     0.05569    |     0.00197    |     0.09845    |

---

> > ### Comment · Reviewer_hP3N · 2023-11-22
> >
> > I appreciate the authors' detailed response and their efforts to address the points raised in my initial review. Consequently, I have decided to change my rating accordingly to "marginally below the acceptance threshold".

---

> > > ### Author Response · Authors · 2023-11-23
> > >
> > > Dear Reviewer,
> > >
> > > Thank you very much for your positive decision and thorough review. Your valuable advice will support us in the further development of the paper.

---

### Meta-Review · Area_Chair_LzDr · 2023-12-04

**Metareview:**

This paper addresses an important task of detecting changes in time-varying HMMs. However, it does not make a clearly substantial methodological improvement over existing approaches. The reviewers are negative or borderline at best and provide some constructive comments that I encourage the authors to incorporate in a stronger revision. The overall idea is nice but the paper as it stands does not make clear why it is better than competitors. Given their study of HMMs using a variational approach, they should cite Emily Fox's paper on SVI for HMMs as well as related literature; there was criticism of the empirical section not making a case for improvement over existing work

**Justification For Why Not Higher Score:**

Negative reviews

**Justification For Why Not Lower Score:**

n/a

---

### Decision · Program_Chairs · 2024-01-16

Reject